METHODS AND RESOURCES

# Efficient and reliable spike sorting from neural recordings with UMAP-based unsupervised nonlinear dimensionality reduction

Daniel Suárez-Barrera[1], Lucas Bayones[1], Norberto Encinas-Rodríguez[1], Sergio Parra[1,2], Viktor Monroy[1], Sebastián Pujalte[1], Bernardo Andrade-Ortega[1], Héctor Díaz[1], Manuel Alvarez[1], Antonio Zainos[1], Alessio Franci [2,3]*, Román Rossi-Pool [1,4]*

**1** Instituto de Fisiología Celular—Neurociencias, Universidad Nacional Autónoma de México, Mexico City, Mexico, **2** Montefiore Institute, University of Liège, Liège, Belgique, **3** WEL Research Institute, Wavre, Belgique, **4** Centro de Ciencias de la Complejidad, Universidad Nacional Autónoma de México, Mexico City, Mexico

☯ These authors contributed equally to this work.
* alessio.franci83@gmail.com (AF); romanr@ifc.unam.mx (RR-P)

## Abstract

Spike sorting is one of the cornerstones of extracellular electrophysiology. By leveraging advanced signal processing and data analysis techniques, spike sorting makes it possible to detect, isolate, and map single neuron spiking activity from both in vivo and in vitro extracellular electrophysiological recordings. A crucial step of any spike sorting pipeline is to reduce the dimensionality of the recorded spike waveform data. Reducing the dimensionality of the processed data is a near-universal practice, fundamentally motivated by the use of clustering algorithms responsible to detect, isolate, and sort the recorded putative neurons. In this paper, we propose and illustrate on both synthetic and experimental data that employing the nonlinear dimensionality reduction technique Uniform Manifold Approximation and Projection (UMAP) can drastically improve the performance, efficiency, robustness, and scalability of spike sorting pipelines without increasing their computational cost. We show how replacing the linear or ad hoc, expert-defined, supervised nonlinear dimensionality reduction methods commonly used in spike sorting pipelines by the unsupervised, mathematically grounded, nonlinear dimensionality reduction method provided by UMAP drastically increases the number of correctly sorted neurons, makes the identification of quieter, seldom spiking neurons more reliable, enables deeper and more precise explorations and analysis of the neural code, and paves new ways toward more efficient and end-to-end automatable spike sorting pipelines of large-scale extracellular neural recording as those produced by high-density multielectrode arrays.

**Data availability statement:** The full code used to implement the spike-sorting pipeline is available on GitHub: https://doi.org/10.5281/zenodo.17575203. The datasets used in this study are publicly accessible. Simulated datasets used in Figs 2 and S3 can be found in: https://doi.org/10.25392/leicester.data.11897595.v1. Data from extracellular recordings during cognitive tasks—used to generate Figs 3, 4, S1, S4, and S5—are available in: https://doi.org/10.5281/zenodo.17546569. The datasets used to compute accumulated neuronal activity during the foreperiod, shown in Fig 4, are provided in: https://doi.org/10.5281/zenodo.17569100. Multi-electrode array (MEA) recordings, which support the analyses in Figs 5, S6, S7, and S8, are available in: https://doi.org/10.5281/zenodo.1205233. The dataset containing simultaneous intracellular and extracellular recordings from the hippocampus, used in Fig 6, can be found in: https://doi.org/10.6080/K02Z13FP.

**Funding:** This work was supported by grants PAPIIT-IN210819, PAPIIT-IN205022 and IN203825 (to R. R. -P.) from the Dirección de Asuntos del Personal Académico de la Universidad Nacional Autónoma de México and CONAHCYT-319347 (to R. R. -P.) from Consejo Nacional de Ciencia y Tecnología; SECIHTI-CBF-2025-I-2054 (to R. R. -P.) from Secretaria de Ciencia, Humanidades, Tecnología e Innovación; IBRO Early Career Award 2022 (to R. R. -P.) from International Brain Research Association. L.B. is a postdoctoral researcher (Postdoctoral fellowship CONACYT-838783). The funders had no role in study design, data collection and analysis, decision to publish, or preparation of the manuscript.

**Competing interests:** The authors have declared that no competing interests exist.

**Abbreviations:** DPC, dorsal premotor cortex; DT, detection task; DWT, discrete wavelet transform; GT, ground truth; HDBSCAN, hierarchical density-based spatial clustering application with noise; MEA, multielectrode array; PC, principal component; PCA, principal component analysis; SC, SpyKING CIRCUS; SNR, signal-to-noise ratio; S2, Secondary Somatosensory Cortex; TICT, time interval comparison task; UMAP, Uniform Manifold Approximation and Projection; VPC, ventral premotor cortex.

## Introduction

Accurately sorting individual neuronal spikes from large-scale recordings is key to understanding how neural activity encodes information in cognitive and sensory functions [1–3], tracing the flow of information across neural networks [4], and examining communication between population subspaces [5]. High-density extracellular electrode array technologies, such as fully integrated silicon probes [6], are enabling the simultaneous recording of hundreds to thousands of neurons [7–9]. Despite this progress, a vast majority of spike sorting methods rely heavily on manual parameter tuning to ensure high sorting accuracy. The need for fast, automatic, and accurate spike sorting methods is now stronger than ever.

To ensure efficient clustering of spiking waveforms, a necessary step in most spike sorting pipelines is to reduce the dimensionality of the processed data. Traditional approaches often employ linear dimensionality reduction techniques, such as principal component analysis (PCA) (e.g., Herding Spikes [10], Klusta [11], Tridesclous, YASS [12], SpyKING CIRCUS (SC) [13], SpikeDeep [14], Perceptron [15]), or Wavelet transform-based methods (e.g., WaveClus [16], WimSorting [17]), supplemented by ad hoc nonlinear metrics (i.e., expert-defined features such as energy or peak-to-peak measures) [16,18]. Existing approaches to spike sorting that include a nonlinear data dimensionality reduction step are usually based on deep-learning methods, like autoencoders and convolutional perceptrons (see, e.g., [19–21], and [15]). Finally, powerful toolkits like Kilosort [22,23] and algorithms tailored to large-scale electrode arrays [13] are constantly being developed to address the problem of achieving fast and accurate spike sorting from hundreds of spatially correlated extracellular recordings.

The dimensionality reduction of spiking data in traditional threshold-based sorting pipelines is critical, and is typically based on linear methods, ad hoc nonlinear metrics, or deep-learning methods. Template-matching methods like Kilosort4 [22,23] have also recently demonstrated high performance by incorporating nonlinear graph-based clustering (e.g., on nearest-neighbor graphs). We do not directly compare to such methods because our focus here is on improving basic threshold-based spike sorting pipelines, i.e., in a way that is agnostic to specific characteristics of spike waveform. Within this context, we show that the Uniform Manifold Approximation and Projection (UMAP) algorithm for unsupervised nonlinear dimensionality reduction drastically increases sorting performance. We also provide rigorous topological arguments to explain this performance gap, particularly for low-firing-rate neurons. Most dimensionality reduction methods lack rigorous, mathematically grounded guarantees that fundamental geometric and topological properties of the original data, and most importantly, the existence of data point clusters, like those associated with spikes from different neurons, are preserved in the reduced data.

The UMAP algorithm [24,25] has recently been introduced to perform data dimensionality reduction with rigorous guarantees that the reduced data preserves key geometric and topological properties of the original data. By preserving and revealing in low-dimensional projections geometric and topological properties of high-dimensional

datasets, UMAP has already proved successful in uncovering the low-dimensional geometry of neuronal population responses [26], in exploring firing rate dynamics and information coding in neural populations [27,28] and in efficiently identifying and classifying cell types [29–32]. In a broader biological context, UMAP has recently been applied to large-scale single-cell datasets, enabling the characterization of immune cell populations and their age- or infection-related dynamics [33], as well as the visualization of developmental trajectories and perturbation responses in transcriptomic models [34–36]. In addition, UMAP requires minimal parameter tuning, is robust to noise and outliers, and scales efficiently to large datasets. All the features and properties of UMAP-based dimensionality reduction make it a good alternative for reducing the dimension of large spiking data in virtually any spike sorting pipelines. By reducing data dimension, UMAP automatically identifies nonlinear geometric structures that serve as automatically-discovered geometric criteria to distinguish spike waveforms with high accuracy, while requiring little to no user intervention. Moreover, UMAP's ability to preserve the data's underlying topological structure regardless of local point density ensures that sparsely represented spike waveforms—such as those from neurons with low firing rates—can efficiently be identified and separated from densely represented waveforms originating from neurons with high firing rates. In contrast, methods that rely on density or variance metrics [13,16] necessarily struggle to isolate low-firing rate neurons. Consequently, many conventional sorting methods tend to overlook quieter neurons, leading to a significant loss of potentially valuable spiking information [37–39].

All the mentioned features of the UMAP algorithm are particularly relevant when dealing with hundreds or thousands of signals measured in a single experiment. As the number of electrodes in an array increases, methods that require parameter hand-tuning (specifying the number of sought spike clusters) or extensive manual curation for guaranteed performance become increasingly unsuitable. With dense multielectrode arrays (MEAs) or multisite recording, it is possible to record from hundreds or even thousands of electrodes simultaneously [7–9]. Therefore, an unsupervised and efficient sorting method is crucial for isolating single-neuron responses, including quiet ones, from these massive databases.

In this study, we introduce a novel spike-sorting pipeline based on UMAP [24]. We apply this pipeline to identify single-neuron responses in several datasets, including synthetic data [16,40], electrophysiological data with an intracellular ground truth (GT) [13,41–43], and in vivo recordings during cognitive tasks [44–48]. Our method shows significant improvements in accuracy compared to existing spike sorting methods such as [49]. The proposed spike sorting method particularly stands out for its robustness to even large heterogeneities in the recorded neuron firing rates. UMAP-based spike sorting is capable to robustly isolate and identify low-firing rate, seldom spiking neurons. Conversely, spike sorting pipelines based on linear dimensionality reduction methods or expert-defined metrics, like peak-to-peak amplitude, temporal width, or axis length in the phase space, just to name a few, tend to either dilute the spike waveforms from low-firing rate neurons into the clusters associated with high-firing frequency ones or to simply ignore them. Furthermore, the unsupervised nonlinear nature of the UMAP algorithm allows it to isolate neurons with spike waveforms that look indistinguishable when looked at through more classical dimensionality reduction methods, based on PCA or Wavelet transform, and that can be challenging to distinguish even through expert-defined metrics. To perform unsupervised sorting of recorded cells from UMAP projections, we employ hierarchical density-based spatial clustering application with noise (HDBSCAN) [50,51]. HDBSCAN is a clustering algorithm that does not require the pre-specification of the number of clusters and can automatically detect data points that cannot be clustered by labeling them as "noise."

We show the superior performance of the UMAP-based spike sorting pipeline using minimal parameter tuning in a variety of settings [52]. The proposed pipeline builds upon but also drastically improves existing spike sorting techniques and technologies [6,13,22,49]. Crucially, we highlight the ability of UMAP-based spike sorting to identify more neurons, particularly low-firing rate ones, as compared to previous methods, which in turn enable more accurate analysis of the information encoded in spikes both at the single neuron and population level. To summarize, we present a novel unsupervised, data-efficient, and high-performance pipeline grounded on UMAP, a nonlinear dimensionality reduction method, to sort individual neuron spikes from both standard and high-density recordings.

## Results

### UMAP-based spike sorting versus traditional pipelines

Fig 1 provides an overview of the proposed UMAP-based spike sorting pipeline, highlighting both its commonalities with existing methods and the unique steps that distinguish it. The four initial steps—high-pass filtering (Fig 1A), spike detection (Fig 1B), waveform alignment (Fig 1C), and representation of the spike waveforms as points in a high-dimensional vector space (Fig 1D)—are standard in nearly all spike sorting workflows. These steps yield a set of spike waveforms, each of which is represented by a vector in $R^n$ (the $n$-dimensional real vector space), where $n$ corresponds to the number of samples in each waveform and the vector components are the recorded samples.

The basic assumption underlying spike sorting is that waveforms from the same neuron are similar, while waveforms from different neurons exhibit distinctive features. Consequently, points in $R^n$ corresponding to spikes from the same neuron should cluster together, whereas points corresponding to spikes from different neurons should remain separated in well-distinguished clusters. The basic goal of a spike sorting algorithm is to efficiently identify the number and identity of such clusters in spike waveform data.

However, clustering in high-dimensional spaces is usually ineffective due to the "curse of dimensionality" [53]. This phenomenon leads to the concentration of measure, where standard metrics (e.g., Euclidean distance) fail to reliably distinguish data points, resulting in unreliable clustering. A possible solution to this problem is to use graph clustering, as done in Kilosort4 [22]. Such an approach is not dissimilar to the one proposed here: reducing data dimensionality using graph-theoretical methods like UMAP and then cluster the low-dimensional projections with standard clustering algorithm. Our approach, however, explicitly separates the dimensionality reduction step from the final clustering, allowing for a more flexible and robust pipeline. Because spike waveforms tend to lie on low-dimensional manifolds [54] over which clustering is feasible, dimensionality reduction remains a critical and near-universal step in threshold-based spike-sorting pipelines. To the best of the authors' knowledge, most existing threshold-based spike sorting methods perform dimensionality reduction either using linear projections (PCA, Wavelet decomposition) or fixed, expert-defined, nonlinear projections (Fig 1E and 1F). Although UMAP has already been used for visualization [55] and in certain specialized spike sorting contexts [56], its integration as the core dimensionality reduction engine in a general-purpose spike sorting pipeline remains largely underexplored. Our approach, by contrast, employs UMAP—a fully unsupervised, nonlinear algorithm that constructs a nearest-neighbor graph to preserve the essential topological and geometrical properties of the processed data [24,30,35]—to reduce the dimensionality of spiking data (Fig 1G–1I).

A common challenge in applying dimensionality reduction to spiking data is that waveforms from different neurons may collapse into a single spurious cluster in the reduced space. Methods such as PCA (Fig 1E and 1F), Wavelet decomposition [16], or expert-defined metrics [23], tend to struggle with complex, nonlinear boundaries between clusters. However, the existence of clusters within a dataset is a topological property that can be formalized using the concept of connected components of a topological manifold [57]. UMAP-based dimensionality reduction can exploit the topological nature of clustering by preserving the topological features of the high-dimensional dataset [24,25] in the low-dimensional projection. It does so in two steps:

1. Uniform Manifold Approximation (UMA), in which the nonlinear structure of the data is encoded into a fuzzy simplicial complex (Fig 1G).

2. Projection (P), in which the fuzzy simplicial complex is embedded into a low-dimensional space (Fig 1H).

Because UMAP preserves clusters through its underlying topological machinery, traditional clustering algorithms can effectively discern putative neurons within the projected data set (Fig 1I). That is, the theoretical guarantees provided by UMAP enhance the prospects of splitting more neurons than feature-based approaches. Spike sorting using UMAP alone, for example, can split all three example neurons in Fig 1F and 1I.

 

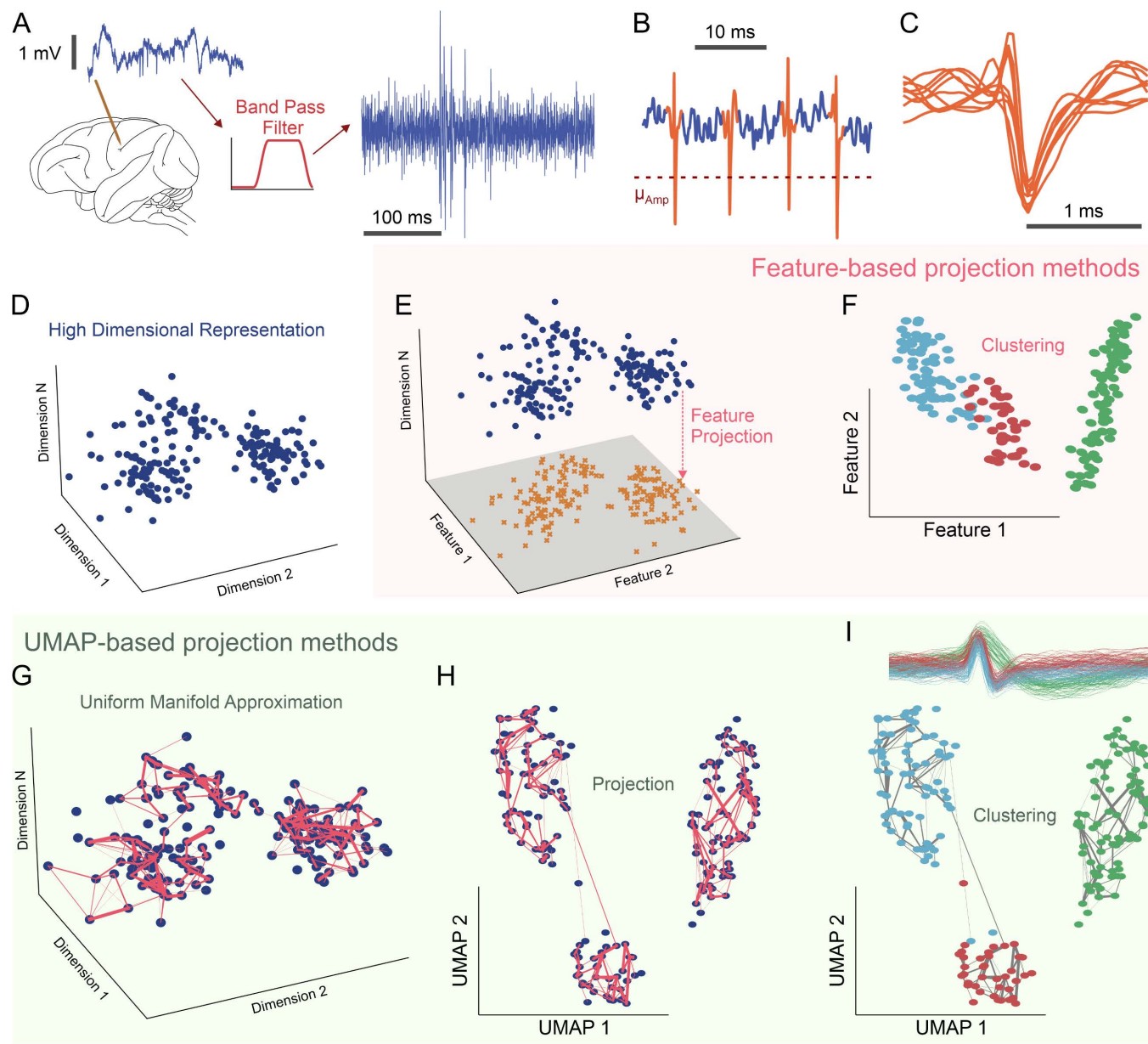

**Fig 1. Comparison of feature-based and Uniform Manifold Approximation and Projection (UMAP)-based spike sorting for neuron classification. (A)** High-pass filtering (500 Hz–2 kHz) removes low-frequency noise from extracellular signals. **(B)** Thresholding the pronounced deflections in the filtered data identifies spikes. **(C)** Windows around each spike, centered on the trough, are isolated to standardize waveform comparisons. **(D)** Each spike waveform becomes a point in a high-dimensional space (time vs. voltage). **(E)** Feature-based sorting relies on linear dimensionality reduction methods (e.g., principal component analysis, wavelets), yet can be thrown off by spikes from different neurons. **(F)** Clustering in this simplified feature space may misclassify neurons due to these linear constraints. **(G)** UMAP-based sorting, however, employs a nonlinear approach that preserves both local and global structure. **(H)** This UMAP projection keeps distinct clusters and neuron-specific features more intact. **(I)** Clustering in the UMAP-projected space better distinguishes individual neurons (shown as distinct colored clusters), reducing merging errors seen with feature-based methods.

For the clustering step, we employed HDBSCAN [51], an unsupervised algorithm that automatically determines the number of clusters, assesses their statistical quality, and labels outliers as noise (pink beads in S1 Fig). S1A–S1D Fig compares HDBSCAN's performance against K-means, and Gaussian Mixture models after UMAP-based dimensionality reduction [52]. Both K-means and Gaussian Mixture require to specify in advance the number of clusters to be detected and lack mechanisms to label outliers as noise. Such conditions normally result in misclassification, particularly for non-convex cluster geometries (for example, electrode A illustrated in S1 Fig, and they also make it harder to separate larger clusters (like electrode B in S1 Fig). In contrast, HDBSCAN's ability to isolate clusters while ignoring outliers provides greater accuracy and consistency in spike sorting results.

These results highlight the limitations of traditional clustering algorithms in identifying the best number of clusters, handling noisy data, and managing highly complex cluster shapes among data points. By integrating UMAP's nonlinear dimensionality reduction technique with HDBSCAN's adaptive clustering framework, the proposed methodology is a more effective and trustworthy approach to neuron separation. The subsequent sections illustrate the competency of this pipeline under varying situations.

## Spike sorting performance metrics

For the analysis presented in the following sections, we use standard metrics to quantify spike sorting performance, particularly when a GT neuron is available (S2 Fig). GT neurons are straightforward to define in synthetic datasets; in experimental datasets, they are usually derived from simultaneous intracellular and extracellular recordings, enabling exact labeling of the GT neuron spikes ($S_{GT}$).

When comparing a sorted spike collection ($S_i$) against the ground truth ($S_{GT}$), we utilize three standard classification metrics (see Methods for detailed definitions):

1. Precision (P): The fraction of spikes in the sorted cluster ($S_i$) that actually belong to the GT neuron. High precision indicates low contamination by spikes from other neurons (low false positives).

2. Recall (R): The fraction of spikes from the GT neuron ($S_{GT}$) that were correctly captured by the sorted cluster ($S_i$). High recall indicates few missed spikes (low false negatives).

3. F1 score: The harmonic mean of precision and recall. It provides a single, balanced measure of sorting quality that accounts for both contamination and losses. $F1 = \frac{2*(P*R)}{(P+R)}$.

A high F1 score requires both precision and recall being high (the optimal case in S2D Fig; F1~1). If recall is high but precision is low, the sorted unit is significantly contaminated (High Contamination case, S2B Fig). If precision is high but recall is low, the sorted unit misses a significant fraction of the GT spikes (High Spike Loss case, S2C Fig).

In scenarios involving multi-electrode recordings, we may analyze the consistency of sorting across channels or compare two sorted spike trains, $S_i$ and $S_j$, without a GT reference. In this case, we define the inclusion index ($I_{i,j}$) as the percentage of spikes in $S_i$ that also appear in $S_j$. We can construct an Inclusion Index Matrix M with off-diagonal elements $M_{i,j} = I_{i,j}$ (S2E Fig). The matrix is generally asymmetric. When a GT unit is present, the corresponding row and column of the Inclusion Matrix correspond to recall and precision values, respectively (as shown in the matrices in Figs 6 and S2A–S2D).

In the following sections, we will rely on the F1 score, precision, and recall to quantify spike sorting performance under a variety of experimental and synthetic conditions.

## UMAP-based spike sorting achieves superior and robust performance

A common challenge in spike sorting arises when the distance between the recording electrode and the recorded neurons increases. Under these conditions, spike waveforms from different neurons begin to overlap, and background noise

becomes more pronounced. To assess the robustness of UMAP-based spike sorting in such scenarios, we used a synthetic dataset created by the Quian Quiroga group [16,40].

Fig 2 compares the performance of the proposed UMAP-based method in sorting a complex mixture of spike waveforms from three synthetic GT neurons (inset Fig 2A) against PCA- and Wavelet-based methods. In each case, HDBSCAN was used to cluster the low-dimensional projections. Notably, the 2D UMAP projection (Fig 2A, left panel) sharply separates the waveforms into three distinct clusters that overlap well with the three GT neurons data points. Conversely, Wavelet (middle panel) and PCA (right panel) projections hardly reveal any well-separated cluster. Although adding a third dimension (S3A Fig for PCA and S3B Fig for Wavelet) can sometimes improve separability, it also complicates visualization, making it harder for researchers to interpret the results—an issue that persists in other datasets (S3A and S3B Fig, right panels). The UMAP method is not affected by these issues, by achieving clear and well-sorted clustering of the data points in just two dimensions, thus preserving ease of visualization and interpretation. In addition, UMAP-based F1 score (Fig 2D) is remarkably insensitive to the used projection dimension ($F1_{UMAP,GT} = 0.83$). By contrast, the F1 scores of Wavelet- ($F1_{WL,GT} = 0.68$) and PCA-based ($F1_{PCA,GT} = 0.58$) sorting exhibits noticeable fluctuation as the used projection dimension is varied, which makes it virtually impossible to determine for these two methods an optimal projection dimension. UMAP-based F1 score is also remarkably robust to more complex and challenging perturbations on the recorded

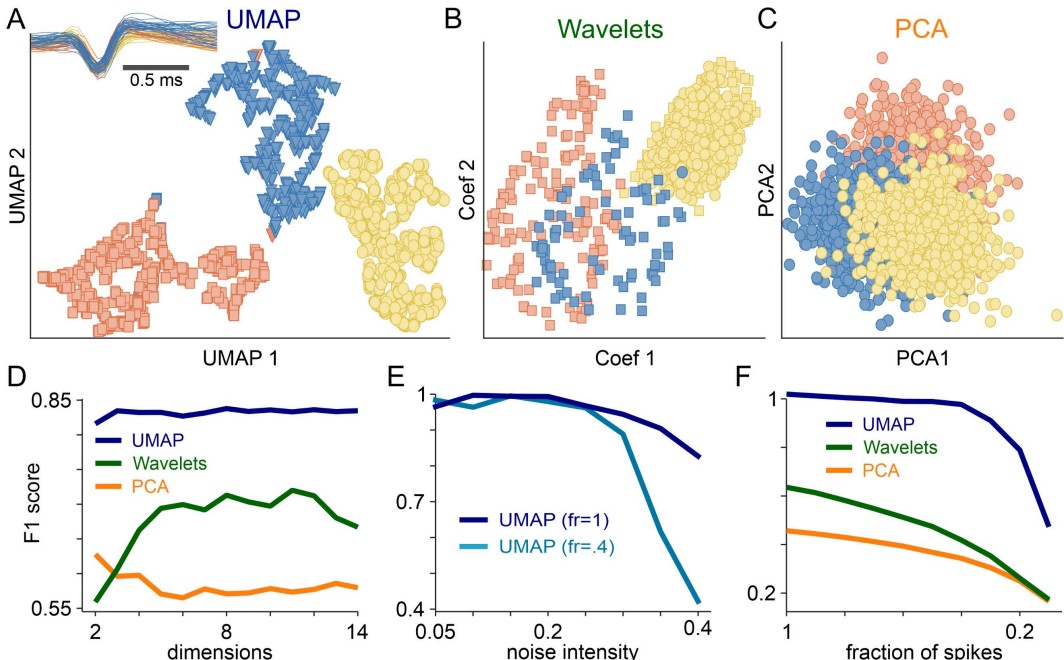

**Fig 2. Comparison of Uniform Manifold Approximation and Projection (UMAP)-based and feature-based spike sorting on synthetic data. (A)** In a simulated dataset containing three different neurons, UMAP-based sorting forms three well-separated clusters with distinct shapes and colors, indicating strong accuracy and minimal overlap. **(B)** Applying wavelet sorting to the same dataset merges the spikes into just two clusters that exhibit noticeable overlap and weak separation, as illustrated in both 3D (top) and 2D (bottom) views. **(C)** Principal component analysis (PCA)-based sorting isolates only a single cluster with poor separation, underscoring PCA's difficulty with complex spike distributions. **(D)** By contrast, UMAP spike sorting consistently surpasses both wavelet and PCA methods, particularly in higher-dimensional spaces, as measured by the F1 score. **(E)** UMAP-based sorting also proves robust against noise perturbations at various spike dilution levels (dark trace; firing rate [fr] at 100% vs. light blue trace; fr diluted at 40%). Even under increased noise and cluster dilution, it preserves high F1 scores. **(F)** Moreover, UMAP-based sorting (dark blue) remains accurate despite data loss, achieving high F1 scores even when a large fraction of spikes (x-axis) is removed from one cluster, whereas wavelet (green) and PCA (orange) methods experience a significant decline in performance. The synthetic data used to generate this figure are publicly available at [40], and the code for performing the analyses is available at [52].

waveforms, like drifting, caused by relative movement between neurons and electrodes, and bursting, which involves high-frequency firing of action potentials (S3C Fig, left and right). In contrast, Wavelet- and PCA-based methods show significant performance declines when faced with the same perturbations.

Because noise is a critical factor in neural spike sorting, we explored how UMAP handles increasing noise levels ($\eta$ ranging from 0.05 to 0.2, where $\eta$ represents the standard deviation of the background noise relative to the amplitude of the spike waveforms [16]) added to three synthetic spike waveforms from [16] (S3D Fig), and compare it against PCA and Wavelet decomposition (S3E Fig, right, middle, and left panels, respectively). Again, UMAP-based sorting is remarkably stable across a wide range of projection dimensions, and it systematically outperforms both PCA- and Wavelet-based sorting as $\eta$ increases. Even at the highest noise level ($\eta = 0.2$), UMAP achieves an F1 score close to 0.6. Finally, to test the ability of UMAP-based sorting to detect "silent" neurons, we randomly removed an increasing fraction of the data points from one of the synthetic GT neuron clusters (the red one, as per Fig 2A). Crucially, in this synthetic dataset, all the neurons have a 20 Hz constant rate, and then after random spike removal, spiking activity is not constant anymore. Remarkably, UMAP still ensured almost perfect (~1) F1 score when only 40% of the cluster data points were retained, and still very good (~0.8) F1 score even when only 20% of the data points were retained. This highlights UMAP's ability to effectively isolated spikes from neurons with low firing rates, which are often key encoders of a task parameters due to their low variability [38,58]. Conversely, the already suboptimal F1 score of both PCA- and Wavelet-based methods dropped much more rapidly as the fraction of removed spikes was increased. Together with its robustness to noise, these findings highlight UMAP's ability to reliably sort low-firing neurons, which suggests it as a powerful spike sorting tool in challenging experimental conditions.

## The cost of overlooking low-firing-rate neurons in neural spike sorting

In Figs 3, S4, and S5, we illustrate how merging spikes from different neurons [46] into a single spurious multiunit entity can severely distort and be detrimental to the analysis of neural information encoding [4,27,45,59]. First and foremost, each multiunit entity created by merging spikes from distinct neurons inevitably exhibits a (potentially much) higher firing rate than the individual units, i.e., the sum of their firing rates. As a result, the average firing rate can be exaggerated, and this can cause significant errors in later analyses. Moreover, neurons with low firing rates are particularly likely to be missed. Their few spikes can be lost in high-firing-rate multiunit clusters unless appropriate dimensionality reduction techniques are used to retain their distinctiveness. Therefore, the potentially important information signaled by low-firing-rate neurons can be buried in the noise of high-firing-rate multiunit clusters. Figs 3A, S4A, and S5A sketch this fundamental problem. While two isolated units might show clear and distinct task parameters and condition encoding (as sketched by well-separated light to dark blue firing rate traces), merging them into a single multiunit entity can average out and thus miss such encoding (represented by overlapping rate traces). This simple phenomenon might erroneously lead to discarding the spurious multiunit activity spikes as irrelevant, thus losing potentially precious and important pieces of neural code.

Figs 3B–3E, S4B–S4E, and S5B–S5E provide different experimental examples of the detrimental merging of two single units into a spurious one. Figs 3B and S4B illustrate the course of a time interval comparison task (TICT) [45,46] in which animals compare the duration of two intervals (Int1 and Int2, ranging from 400 ms to 2000 ms) and report which one is longer. During the 2-second working memory period between the intervals, the animal must retain Int1-related information in order to compare it with Int2. The anatomical diagram in the same panel shows the location of the dorsal premotor cortex (DPC), where the neurons in Figs 3C–3E and S4C–S4E were recorded from. Figs 3C and S4C show the raster plots (top) and firing rates (middle), aligned to the start of the delay period, of a multiunit entity created by merging two putative neurons identified through the UMAP-based pipeline, but that were combined in a single (higher firing rate) neuron when the PCA-based method was used. The activity of the two individual neurons is shown in Figs 3D, 3E, S4D, and S4E. Notably, the putative neuron in Fig 3D has a relatively lower firing rate but carries significantly more information about Int1 during the delay period, as revealed by mutual information analyses (bottom panels in Figs 3C–3E and S4C–S4E). The multiunit

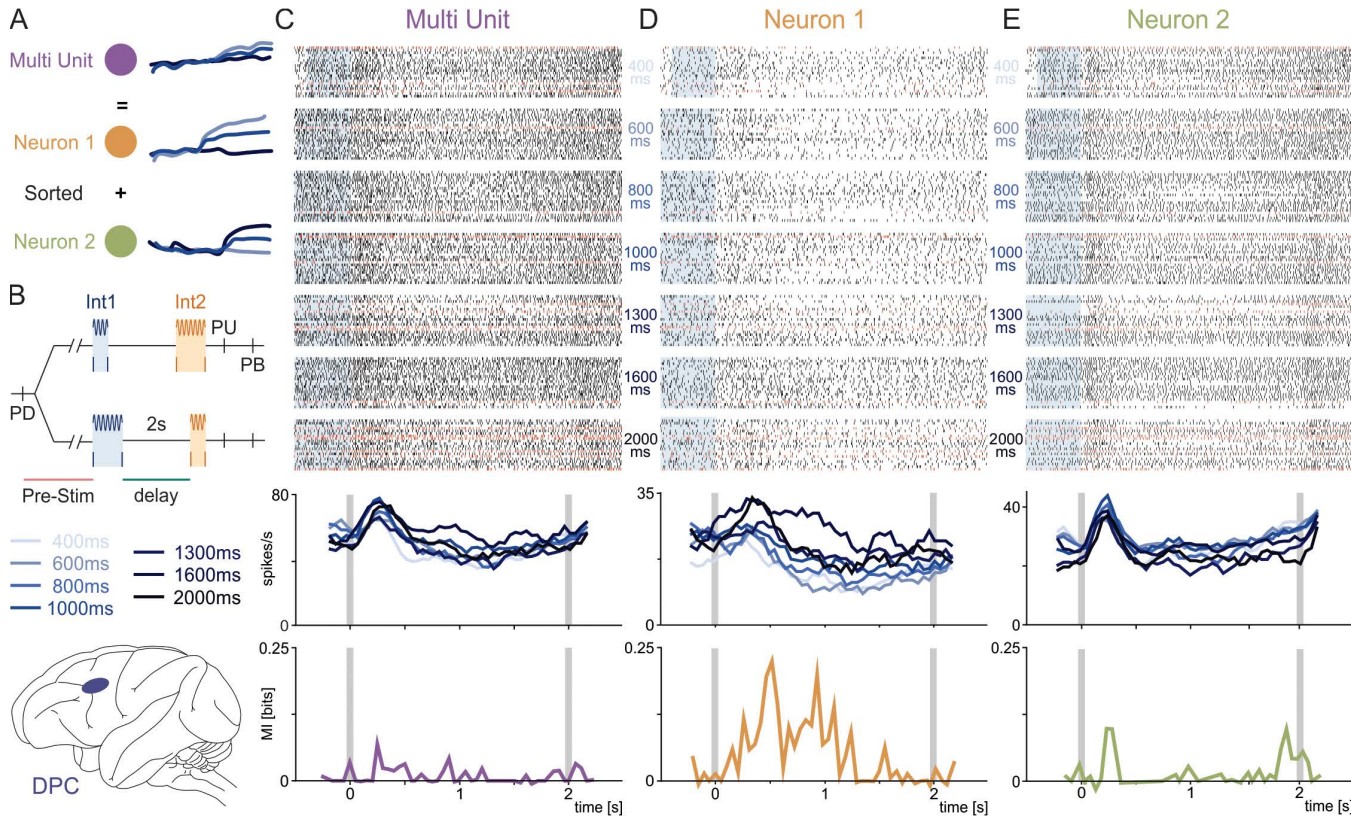

**Fig 3. Impact of multiunit merging on neural information encoding.** Multiunit merging, a common artifact in spike sorting, combines spikes from multiple neurons into a single "putative neuron." This inflates firing rates—particularly masking low firing rate neurons—and obscures their distinct information-encoding properties. **(A)** Schematic illustrating how merged activity from two true neurons can appear task-irrelevant, as the combined firing pattern shows little difference between two task conditions (green and orange vs. purple). Proper sorting reveals that each neuron encodes unique information. **(B)** Time Interval Comparison Task (TICT). Monkeys compare two intervals (Int1 and Int2) separated by a 2-second delay, then press one of two buttons to indicate which interval was longer. Int1 (400 to 2000 ms) is shown in shades of blue. Data was recorded in dorsal premotor cortex (DPC, shown in brain cartoon). **(C)** Top: Raster plot of a misclassified multiunit in the DPC, aligned to the end of ($t = 0 =$ start of the delay). Gray rectangles indicate Int1 backpropagated from $t = 0$. Black ticks indicate spikes during correct trials; red ticks indicate spikes during error trials. Bottom: The average firing rate and mutual information fail to show meaningful encoding. **(D)** The first neuron contributes to the multiunit (panel C). Despite a low firing rate, it exhibits a clear positive encoding of Int1 (larger stimulus intensities produce higher firing rates). **(E)** The second neuron shows a strong early sensory response followed by a negative encoding of Int1. Merging these two distinct patterns masks each neuron's specific role in information processing. The multiunit and UMAP sorted neuronal activity used to generate the raster plots and firing rates is publicly available at [46], and the code to compute firing rates and mutual information is available at [52].

activity in Fig 3C carries significantly less information than the combined values of the two isolated neurons, which provides evidence and an experimental example of how merging different neurons into spurious multiunits can erase potentially critical information carried by low-firing-rate neurons. Although the putative neuron in S4D Fig is almost silent most of the time—which is probably a key reason for why it is likely to be missed during sorting—it carries more information about Int1 at the onset of the delay period than both the multiunit entity in S4C Fig and the other putative neuron in S4E Fig.

S5 Fig shows a similar effect in a different task and brain area. S5B Fig outlines the course of a tactile detection task [44,46,59] and shows the location of the ventral premotor cortex (VPC), where the neurons in S5C–S5E Fig were recorded from. In this task, the animals report whether a vibrotactile stimulus (0–24 μm in amplitude) is present or not. S5C Fig displays the multiunit activity obtained by merging the spikes of the two putative neurons in S5D and S5E Fig. The two neurons in S5D and S5E Fig differ substantially in both firing rates and encoding dynamics: the neuron in panel E

persistently maintains a stable encoding during the delay period, while the neuron in panel D responds strongly, but only transiently, to stimulus presentation and its activity rapidly decreases during the delay period. Merging these two putative neurons completely conceals their distinctly different encoding profiles–most critically, it erases the persistent encoding of the lower-firing-rate neuron (E) and diminishes the nuanced, non-monotonic delay activity of the higher-firing-rate neuron (D).

The experimental evidence provided in this section shows that dimensionality reduction methods that are sensitive to data-point density—such as PCA–likely fail to detect and isolate the neurons with lowest firing rates. Their spikes tend indeed to be subsumed into multiunit clusters dominated by higher-firing-rate neurons. By drawing on topological and geometric principles, UMAP-based spike sorting addresses this issue effectively, preserving the unique contribution of low-firing-rate neurons and safeguarding against the loss of valuable information.

**UMAP-based spike sorting improves neural information encoding analysis by enhancing low-firing-rate neurons detection**

In this section, we applied UMAP-based sorting to recordings from three brain areas—DPC, VPC, and secondary somato-sensory cortex (S2)—while animals perform a tactile detection task and TICT [45–47] previously outlined in Figs 3B, S4B, and S5B, respectively. Our objective is to show that isolating low-firing-rate neurons through effective spike sorting not only increases the total number of identified cells but also boosts the amount of information extracted from the data. Figs 3, S4, and S5 report examples of this phenomenon. Fig 4 reveals that the same phenomenon is much more general.

Fig 4A shows the firing-rate cumulative probability distribution of the foreperiod activity (data in [47]) from the putative neural population sorted using either UMAP-based or PCA-based methods from DPC, VPC, and S2 (brain views on Fig 4B) recordings from TICT and tactile detection task. In total, 129 (UMAP) and 96 (PCA) neurons were identified in DPC, 228 (UMAP) and 145 (PCA) in VPC, and 134 (UMAP) and 60 (PCA) in S2. In every dataset, UMAP-based sorting consistently identifies more lower-firing-rate than PCA-based.

Performing this analysis in each brain area using the activity of the tactile detection task [46]—S2 (Fig 4C), VPC (Fig 4D), and DPC (Fig 4E)—we confirmed that UMAP-based sorting consistently detects a larger population of low-firing-rate neurons across all regions (left panels). Furthermore, in all areas, UMAP-based sorting significantly increased the total number of simul-taneously recorded neurons (central panels). This increment in simultaneously recorded neurons count is critical for exploring pairwise spike correlations [1–3]. For instance, in VPC, the number of simultaneously recorded neurons—obtained across 12 recording sessions—rises from roughly 57 to nearly 113, thereby increasing the average number of neuron pairs from about 5 to almost 9. Finally, the mutual information extracted from neuronal spikes about task parameters was consistently higher when using UMAP-based sorting (right panels). This observation is consistent with the enhanced clarity and cluster separation achieved by UMAP-based sorting, highlighted in Fig 3. In short, UMAP-based sorting not only identifies more neurons—particu-larly those with low firing rates—but also enhances the overall information decoded from those cells.

**Comparing UMAP-based sorting and SpyKING CIRCUS in multielectrode array recordings: The role of spatial information**

In this section, we explore the performance and potential of UMAP-based spike sorting on MEA recordings [13,42]. We compare UMAP-based sorting with SC [13], a widely used, high-performing software specifically designed for in vivo and in vitro MEAs that exploits the MEA spatial information redundancy by processing all electrodes simultaneously for more accurate spike sorting. Performance was quantified using the dataset [42] from the original study [13], which includes extracellular recordings from a surface MEA alongside intracellular recordings that define a GT neuron.

We conducted the assessment using two different approaches for the UMAP pipeline. First, we applied UMAP inde-pendently to the data from each electrode. While this approach does not leverage the spatial redundancy of the MEA,

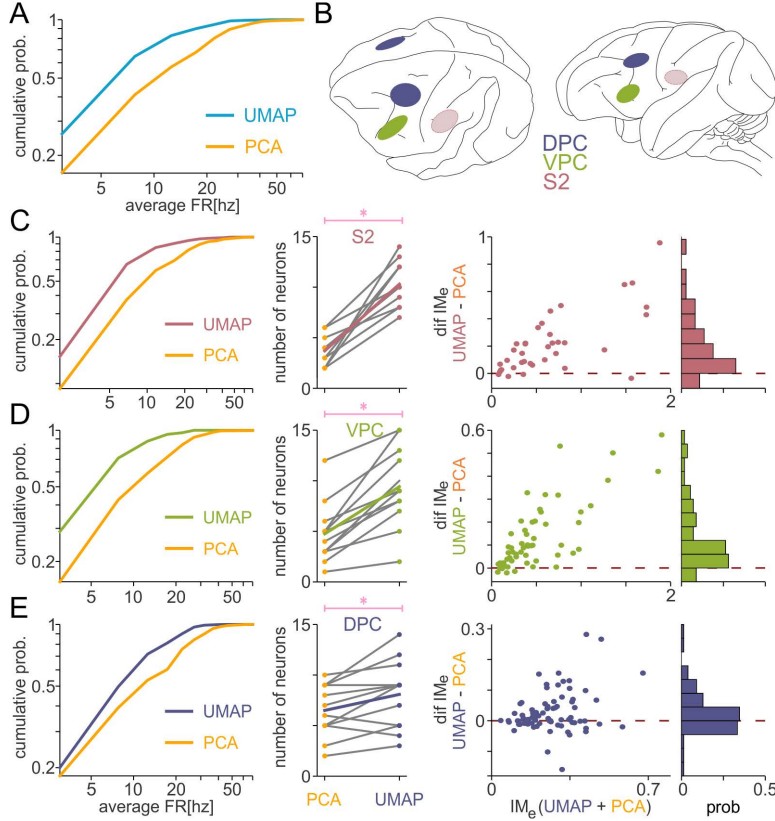

**Fig 4. Uniform Manifold Approximation and Projection (UMAP)-based sorting enhances detection of low-firing rate neurons and boosts mutual information. (A)** Cumulative average firing rate distributions for UMAP-based and principal component analysis (PCA)-based spike sorting reveal that UMAP identifies more low-firing neurons, reflecting greater sensitivity to sparse firing. **(B)** Anatomical maps illustrate three cortical regions: ventral premotor cortex (VPC, green), secondary somatosensory cortex (S2, red), and dorsal premotor cortex (DPC, blue). Dotted lines in the lightly shaded oval above S2 indicate recording sites in deeper cortical layers—areas involved in higher-level processing and decision-making. **(C–E)** Comparisons of PCA and UMAP sorting across these regions. Left: UMAP consistently recovers a larger number of low-firing neurons based on cumulative firing rate distributions: 134 (UMAP) and 60 (PCA) neurons were identified in S2; 228 (UMAP) and 145 (PCA) in VPC; 129 (UMAP) and 96 (PCA) in DPC. Center: It also yields higher overall neuron counts from the same sessions: average number of neuron pairs in S2 from about 4 to almost 10; VPC from about 5 to almost 9; DPC from about 6 to almost 8. Right: UMAP reveals greater mutual information (MI) compared to PCA, indicating stronger information decoding across all brain regions. In summary, UMAP effectively increases neuron recovery, captures low-firing activity, and enhances the information gained from cognitive task data. The neuronal activity recorded during the foreperiod of the cognitive tasks, used for the cumulative distributions, is available at [47]; the single neurons from different sessions used for the mutual information calculations can be found at [46]; and the corresponding analysis code is available at [52].

it allows us to assess the inherent robustness of the dimensionality reduction method to the lower signal-to-noise ratios (SNRs) encountered as the electrode-to-GT neuron distance increases. Here, we identified the sorted unit corresponding to the GT neuron and computed the F1 score retrospectively at each electrode location within a 5×5 grid centered on the GT neuron (Fig 5A–5D). The position of each electrode in the grid is represented by a coordinate $(i, j)$, $i, j = -2, \ldots, 0 \ldots, 2$, with $(0, 0)$ corresponding to the electrode closest to the GT neuron.

Remarkably, UMAP-based sorting applied independently to single channels exhibited a gradual decay in F1 score as the distance increased (Fig 5A and 5B), suggesting that UMAP provides a low-dimensional embedding that is robust to low SNR, even when restricted to only use single-channel data. Fig 5C and 5D summarize these observations by grouping electrodes with similar distances from the GT neuron and plotting how F1 score changes with distance. It becomes evident how F1 score slowly decreases, remaining significantly above zero up to 40 μm away (more examples on S6A Fig). An

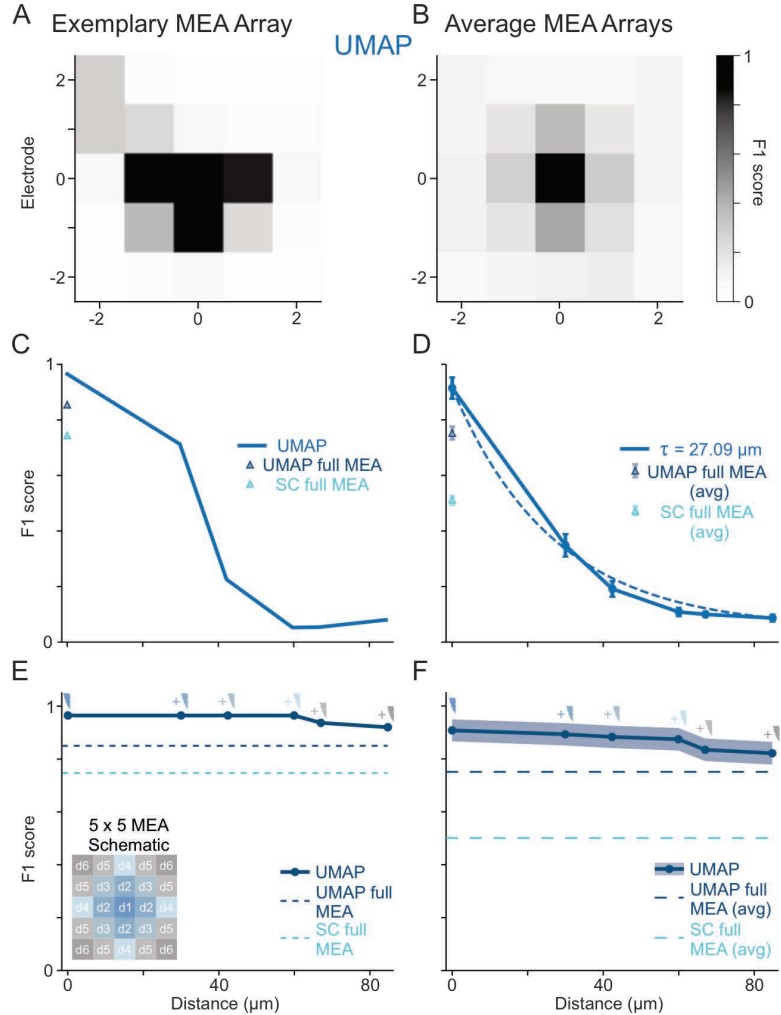

**Fig 5. Spatiotemporal spike sorting on multi-electrode array (MEA) data: Uniform Manifold Approximation and Projection (UMAP) vs. SpyKING CIRCUS (SC).** We compare spatial sorting performance on MEA data using UMAP-based sorting vs. SC, with intracellular recordings as ground truth (GT). Performance is measured by the F1 score (see S2 Fig). **(A, B)** Spatial matrix of F1 scores for an exemplary MEA (A) and average across all MEAs **(B)** using single-channel UMAP-based sorting (applied independently to each). The gradual decline in values with the distance from the electrode indicates that spike detection is still reliable at larger separations. The average F1 score matrix from different MEA recordings shows a similar gradual decrease, emphasizing UMAP's ability to detect signals at larger distances from the GT neuron. **(C)** F1 score vs. distance for the exemplary MEA shown in A. UMAP (blue trace), maintains a good detection performance (F1 score) with distance at even 40 μm far from the central electrode. Triangles indicate the overall F1 score when analyzing the entire MEA signal simultaneously (UMAP simultaneous full MEA, dark blue; SC simultaneous full MEA, light blue). **(D)** Average F1 score vs. distance across all MEAs shows that UMAP-based sorting recovers GT spikes consistently across tens of micrometers. Dashed lines show the exponential fit to F1 values as a function of electrode's distance. **(E)** UMAP geometric analysis (dark blue trace), where the signals of neighboring electrodes (exemplary MEA above) are progressively added before the dimensionality reduction step (small lightning's represent signals of the different electrodes added before each dimensionality reduction instance). Dashed lines represent the overall F1 values shown as triangles in C. **(F)** Average of the geometric analysis employing UMAP across all MEAs. Long dashed lines represent average of overall F1 values using UMAP (dark blue) and SC (light blue) represented with triangles in D. MEA recordings supporting the analyses are available at [42], and the code necessary for the analyses is available at [52].

exponential fit of the data across MEAs in Fig 5D offers a more quantitative view of this smooth spatial decay, yielding a characteristic length constant of τ = 27.09 μm.

Additionally, UMAP-based sorting is robust not only to heterogeneous firing rates and noise but also to signal variations arising from the distance between electrodes and neurons. As shown in S6B and S6C Fig, this method accurately

identifies the same spike across neighboring electrodes (see panel C, middle for a clear example). This performance highlights UMAP's ability to leverage both spatial and temporal information.

Next, to make a fair comparison of the way SC and UMAP-based sorting uses spatial information, we implemented two possible multielectrode versions of the UMAP pipeline. In the first one, we computed F1 score using the stacked signals from all electrodes and compared it with the one obtained with SC (UMAP and SC full MEA, blue and light-blue triangles in Fig 5C and 5D). With this multielectrode implementation, UMAP-based sorting achieved a superior performance as compared to SC, showing its power for spike sorting relying on distance metrics in extremely high dimensions. For the second implementation, we employed a more geometric approach (Fig 5E and 5F). Specifically, spike waveforms from increasingly more distant neighboring electrodes were progressively aggregated prior to the dimensionality reduction step (see schematic on Fig 5E), which should allow UMAP to exploit spatiotemporal correlations the same way SC does. The results demonstrate that UMAP-based sorting performance remains highly stable when integrating spatial information progressively (see S7 Fig for more individual examples), making it a powerful candidate for multielectrode spike sorting.

Finally, to understand the factors limiting performance as the distance from the GT neuron increases, we dissected the F1 score into its components (S8 Fig), that is, precision, the probability that a sorted spike belongs to the GT neuron, and recall, the probability that a GT spike is correctly sorted. Both probabilities exhibit distance-dependent decay. However, recall drops more sharply than precision (S8C and S8F Fig), indicating that lost GT spikes—rather than contamination from spikes of other neurons—are the primary driver of the decrease in F1 score as distance increases. Interestingly, while the overall precision was comparable between the UMAP and SC methods when using all electrodes (S8C Fig, light-blue and cyan triangles, respectively), SC's recall was substantially lower than that of UMAP-based sorting (S8F Fig, yellow and orange triangles, respectively). This indicates that while both methods have similar rates of contamination (false positives), SC tends to miss a significant number of GT spikes that the UMAP approach successfully retains.

### Comparing UMAP-based and feature-based spike sorting in multielectrode recordings

In Fig 6, we applied UMAP-based sorting to extracellular multielectrode recordings obtained in Buzsáki's lab [41]. This dataset utilizes tetrodes [43], which are specifically designed to increase the separability of neural spike waveforms by recording them simultaneously on multiple closely-spaced electrodes. While we demonstrated in the previous section (Fig 5) that the UMAP pipeline effectively integrates multichannel information, here we analyze the recording of each electrode separately. This approach allows us to isolate the intrinsic robustness of the dimensionality reduction methods (UMAP, Wavelet, PCA) when dealing with varying signal quality across the electrodes, independent of spatial integration techniques. This complements the analysis in Fig 5 by further testing the quality of the low-dimensional embedding provided by each method in low-SNR conditions. Also in this database, an intracellular electrode was used to define a GT neuron whose extracellularly recorded spike waveforms are collected in $S_{GT}$. Simultaneously, four extracellular electrodes were placed at varying distances from the GT neuron (Fig 6E). UMAP-based sorting of electrode 1 data revealed four well-separated clusters (Fig 6A) associated with the waveforms in Fig 6D. One of the clusters almost perfectly overlaps with $S_{GT}$, indicating successful identification of the GT neuron. The other three cluster points to three other putative neurons. Fig 6H and 6K show the same data points as Fig 6A and colored according to the same UMAP-based clustering, but projected using Wavelet decomposition and PCA, respectively. As detailed below, it is already clear from these projections that neither Wavelet- nor PCA-based sorting are able to identify and sort the GT neuron. In Fig 6F, data points that are shared with Fig 6A (i.e., spikes recorded on both electrode 1 and 3) are colored according to the same cluster colors as Fig 6A, whereas data points associated with spikes recorded on electrode 3 but not electrode 1 are drawn in purple. Notice that electrode $E_3$ is farther from the GT neuron than $E_1$ (Fig 6E), making it more difficult to sort spikes from the GT neuron, in agreement with Fig 5.

The 1st line of Fig 6B, 6I and 6L, shows the GT neuron raster plot, drawn in green, while the $i$-th line (starting from bottom to top, $i = 1, \ldots, 4$), shows the raster plot of the spike cluster $S_i$ sorted from electrode $E_i$ (either through UMAP-,

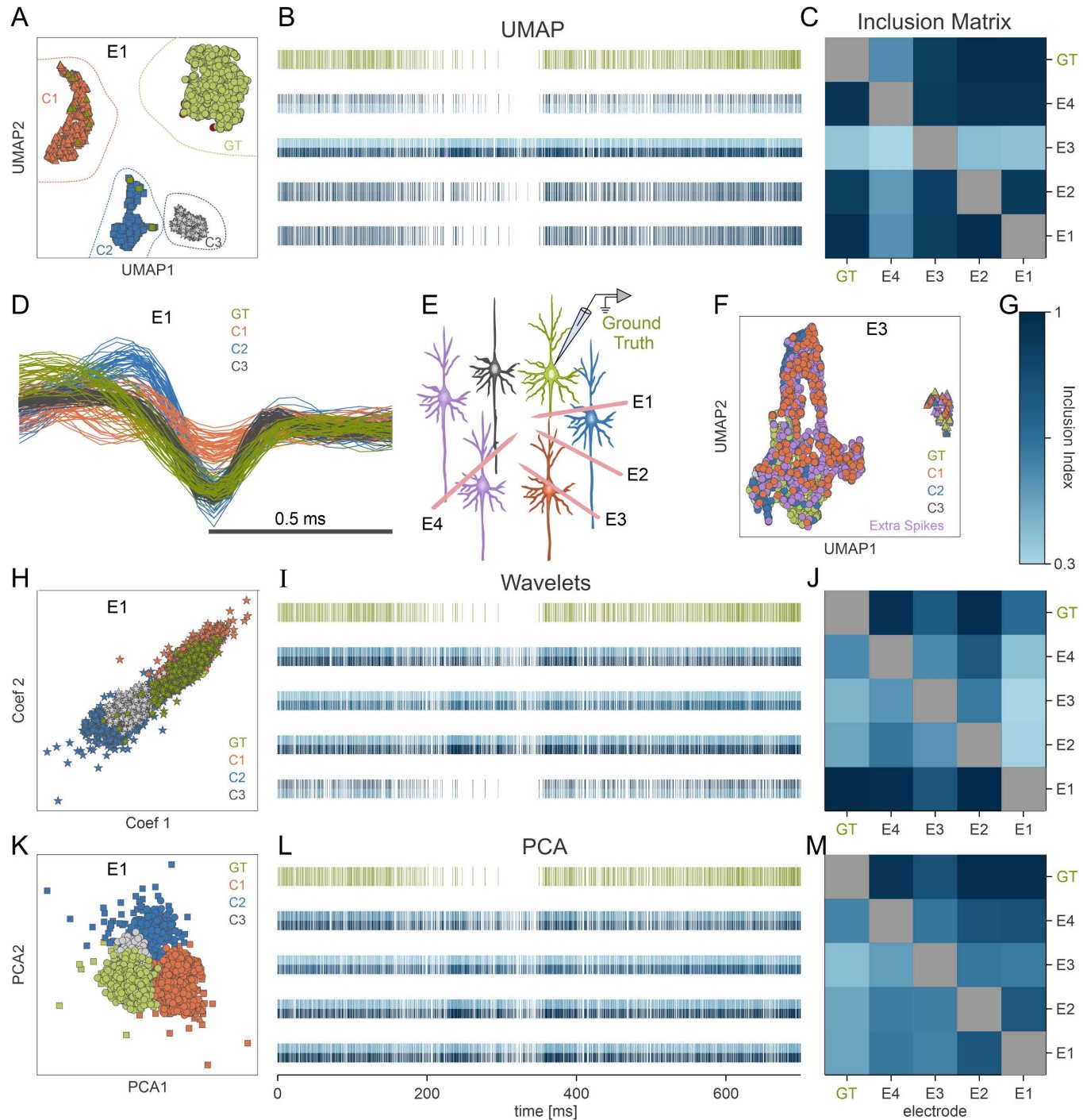

**Fig 6. Comparative analysis of spike sorting using Uniform Manifold Approximation and Projection (UMAP) vs. feature-based methods in in vivo extracellular recordings.** Recordings from Buzsáki's lab [26] include an intracellular electrode serving as ground truth (GT, green) and four extracellular electrodes ($E_1$–$E_4$) placed at different distances from the GT neuron. **(A)** UMAP-based sorting on electrode $E_1$ identifies four clusters (distinct shapes). One cluster ($S_{E_1}$, circular markers) overlaps substantially with the GT neuron, while the other three represent separate putative neurons. **(B)** Raster plots for GT clusters found by UMAP on each extracellular electrode, alongside the GT neuron's raster. Each raster line is horizontally split; the color saturation of the top half represents the precision (percentage of cluster spikes that come from the GT), and the color saturation of the bottom half indicates the recall (percentage of GT spikes captured by that cluster) (see color bar in G). Electrodes $E_1$ and $E_2$ align well with the GT neuron (uniform dark blue), whereas $E_3$'s cluster is contaminated (low precision), and $E_4$'s cluster loses many GT spikes (low recall). **(C)** Inclusion Index matrix $M_{i,j}$ for

UMAP, illustrating overlap of GT spikes across electrode clusters. Darker squares for $E_1$ and $E_2$ reflect accurate capture of GT spikes, while $E_3$ and $E_4$ show contamination and loss, respectively. **(D)** UMAP-sorted waveforms from $E_1$, color-coded to match the clusters in (A). **(E)** Experimental schematic from Buzsáki's lab, showing the intracellular GT neuron and four extracellular electrodes. **(F)** UMAP sorting on electrode $E_3$ highlights significant contamination, with many not-GT spikes mislabeled as GT (shown in a separate color). **(G)** Color bar for the inclusion matrices (C, J, M). **(H–J)** Wavelet-based sorting: (H) projection of spikes from $E_1$ in Wavelet feature space, identifying multiple clusters. (I) Raster plots for the GT clusters on each electrode. The GT neuron's pattern is not well isolated. (J) Inclusion matrix shows low, inconsistent overlap for GT spikes across electrodes. **(K–M)** PCA-based sorting: (K) PCA projection for $E_1$, resulting in broad, overlapping clusters. (L) Raster plots for these GT clusters showing substantial contamination. (M) Inclusion matrix reflects the poor specificity of PCA-based clustering, with larger fractions of non-GT in each putative cluster. The data containing simultaneous intracellular and extracellular recordings from the hippocampus can be found at [43], and the analysis code is available at [52].

Wavelet-, or PCA-based methods, as indicated in the panel titles) that most closely matches (in the sense of maximizing the F1 score) the GT neuron spike cluster $S_{GT}$. Observe that each raster plot is chromatically split into two halves, distinguished by different blue saturation levels, along the horizontal axis of the plot. The two saturation levels represent the sorting performance of the raster spike cluster with respect to two different metrics, namely, precision (top half) and recall (bottom half). Hence, only the raster plots of spike clusters with a high F1 score are drawn in a roughly uniform, highly saturated blue tone.

The cluster obtained from electrode $E_3$ in Fig 6B using UMAP-based sorting exhibits low precision but high recall, suggesting contamination with non-GT spikes (S2B Fig). Conversely, the cluster obtained from electrode $E_4$ exhibits high precision but low recall, indicating a significant loss of GT neuron spikes (S2C Fig). By contrast, the two clusters obtained from electrodes $E_1$ and $E_2$ are characterized by high precision and high recall, thus identifying the GT neuron spikes with high quality (Fig 6A). Fig 6C shows the remaining elements of the Inclusion Index matrix ($M_{i,j}$) that can be used to investigate inclusion relationships between the spike clusters sorted from different electrodes (see S2 Fig).

Fig 6H–6M shows the result of the same analysis but using Wavelet decomposition and PCA, respectively, instead of UMAP for the dimensionality reduction step. As shown in Fig 6H, when only the two largest Wavelet coefficients are used to project the data, all spike waveforms are merged into a single cluster. In spite of this, the utilization of the six most considerable coefficients enhances the separation process, leading to the improved clustering results shown in Fig 6I and 6J; however, none of the clusters derived from the Wavelet method as precisely resembles $S_{GT}$ as those derived through UMAP-based sorting. The cluster obtained from electrode $E_1$ is of the best quality of sorting, although it undergoes a comparatively high loss rate. In contrast, the clusters obtained from electrodes $E_2$, $E_3$, and $E_4$ are of inferior qualities of sorting because they experience a high contamination rate. PCA-based sorting is worse and consistently produces big, highly contaminated clusters of high-firing frequency multiunit activity for all electrodes.

In summary, when analyzing the electrodes independently, only UMAP-based spike sorting reliably separates GT neuron spikes from at least two electrodes. Wavelet-based and PCA-based methods either merged multiple neurons' spikes (high contamination) or failed to capture a significant fraction of the GT neuron's spikes (high loss). These results underscore that UMAP provides a superior low-dimensional embedding compared to PCA and wavelet-based methods, enabling effective sorting even in the challenging, low-SNR conditions encountered as the electrode-to-neuron distance grows, consistent with the findings in Fig 5.

## Discussion

Spike sorting, i.e., the identification and classification of single neuronal spikes from extracellular electrophysiological recordings, is essential to understanding neural codes. Conventional approaches typically use thresholding techniques in conjunction with linear dimensionality reduction methods, e.g., PCA or Wavelet decomposition, followed by clustering [16,17,60]. Although effective for high-amplitude or regularly firing neurons, such linear approaches are ineffective at separating clusters that are divided by faint, nonlinear waveform disparities, particularly for large-scale recordings, thus leading to neuron misclassification and manual sorting of large quantities [6]. In contrast, nonlinear dimensionality reduction

methods—especially UMAP—provide mathematically rigorous embeddings that preserve both global and local topological relationships within spike waveform data [24,25].

While analyzing massive datasets without spike sorting has recently gained popularity for exploring population-level latent dynamics [61,62], critical single-neuron analyses can be severely compromised if sorting is omitted [63]. Vital insight, for example, derived from the relationship between noise correlation and signal correlation, depends largely on properly sorted single-neuron data [1–3]. Moreover, signal correlation must be orthogonal to noise correlation to achieve maximum population-encoded information [3,64]. Neglecting spike sorting risks losing valuable information about single-neuron dynamics, obscuring fundamental properties of neural computation. Our spike-sorting pipeline directly addresses these challenges, leveraging UMAP's nonlinear embeddings to successfully separate clusters based on subtle waveform differences with high precision, outperforming linear methods by a large margin in a range of experimental conditions (Figs 2–4). This capacity not only reduces manual curation but also ensures complete neuron detection, including low-firing units typically omitted.

One important subset of neurons that is usually neglected consists of low-firing or "silent" neurons [38,58]. Conventional approaches are typically not able to identify these neurons because of the low firing rates and low spike amplitudes, which are difficult to discriminate using variance-based clustering [10,65]. In contrast, UMAP's robustness to variability in spike densities provides strong detection and classification of "silent" neurons. Empirical results indicate that UMAP obtains a very high sorting accuracy even under extreme spike loss of up to 80% and that linear methods suffer a drastic reduction in performance. Thus, the application of UMAP for sorting facilitates the detection of more diverse and accurate neuronal populations, which greatly simplifies subsequent neural coding analysis and interpretation in cognitive neuroscience research [1–3,64]. Further, by maximizing both the number and diversity of the detected neurons, UMAP substantially improves studies on neural information encoding. Low-firing neurons, which are conventionally disregarded by default procedures, carry valuable, stable information with finely graded response variability [3,39,64]. Our data for cognitive tasks demonstrate that omission of such neurons substantially distorts firing rate estimates and mutual information estimates of task-relevant variables. By accurately preserving low-firing neurons, UMAP yields higher quality sorted datasets and more valid interpretations of population codes, noise correlations, and task-related neural dynamics [6,7,10].

Another significant advantage of applying UMAP to spike sorting is its remarkable ability to leverage the spatial and temporal features contained in MEA recordings. By implementing a pipeline [52] that integrates features across electrodes prior to dimensionality reduction (geometric analysis), we demonstrated that UMAP performs outstandingly well when capturing spatial features as conventional specialized MEA sorting algorithms that use the full array data (Fig 5). We hypothesize that UMAP's capacity to identify complex, nonlinear structures and optimize its distance in higher dimensions, allows it to exploit subtle spatiotemporal signatures effectively. Furthermore, UMAP demonstrated a superior performance even when analyzing electrodes independently (Figs 5 and 6), highlighting the intrinsic quality of the nonlinear embedding in low-SNR regimes. This robust spatiotemporal performance enables the identification of genuine spikes across a wider area of the array (S6 Fig), improving neuronal identification and the analysis of functional connectivity [66–69]. Besides, UMAP's insensitivity to temporal fluctuations like electrode drifting and neuron bursting (S3C Fig), these advantages confirm its suitability for large-scale MEA, closed-loop experimental paradigms, and real-time applications, where stable and high-quality neural representations are essential [21,70–72].

Computational scalability is yet another key benefit of our UMAP-based pipeline, essential for modern high-density electrophysiological experiments with thousands of neurons recorded simultaneously [6,9]. In contrast to methods like t-SNE or Isomap, which become intractable with large datasets, UMAP scales roughly linearly with dataset's size [24,25]. Our results consistently demonstrate UMAP's higher performance compared to linear methods, making it especially beneficial for real-time or near-real-time spike sorting in closed-loop experimental paradigms. Observably, the stable nonlinear embedding of UMAP, combined with the density-based clustering of HDBSCAN, enables an end-to-end, fully automated spike sorting pipeline. Unlike clustering algorithms such as Gaussian mixture models or K-means, which require

predefined cluster numbers and struggle with non-convex cluster borders, UMAP-HDBSCAN automatically determines optimal clusters and considers outliers as noise. This automation significantly reduces and accelerates the spike sorting pipeline, with minimal manual intervention and maximal reproducibility.

A practical consideration when implementing UMAP is the selection of hyperparameters, most notably the number of neighbors (n_neighbors), which influences the balance between preserving local versus global dataset structures. While this parameter can affect the appearance of the embedding, we found that the sorting results (the clusters identified by HDBSCAN) were robust across a wide range of values. In our analyses, we simply utilized the default UMAP parameters (and in particular n_neighbors = 15; see Methods [52]), achieving high performance without extensive tuning. This insensitivity to parameter choice, combined with HDBSCAN's automatic determination of cluster numbers, underscores the suitability of this pipeline for fully unsupervised, high-throughput analysis.

Relative to other contemporary spike sorting approaches with UMAP, like WaveMAP [31] (used for cell type classification), which involves graph-based clustering, or P-sort [56] (a spike sorting method tailored for the cerebellum), our method utilizes only UMAP along with HDBSCAN clustering. While recent deep-learning methods like SimSort [16,17,60] have demonstrated excellent performance by leveraging extensive training on simulated data, our approach provides a powerful, unsupervised alternative that does not require prior training or assumptions about waveform shapes. This direct and efficient method obviates additional preprocessing or specialized clustering algorithms, thus enhancing computational efficiency, enabling automation, and increasing generalizability to various types of neurons and experimental conditions. Consequently, our method effectively captures a wider array of neurons, substantially improving analyses of population-level neural coding and theoretical insights into neural computation. Future studies could further validate our method's broad applicability, potentially integrating deep learning techniques for a more powerful, comprehensive unsupervised sorting pipeline.

## Methods

### Ethics statement

Neuronal recordings were obtained from S2, VPC, and DPC while monkeys performed the detection task (DT) and the time interval comparison task (TICT). All animal procedures were conducted in accordance with NIH and Society for Neuroscience guidelines. Protocols were approved by the Institutional Animal Care and Use Committee of the Instituto de Fisiología Celular, Universidad Nacional Autónoma de México. Study approval numbers: RRP247-24 for DT and RRP246-24 for TICT.

### Spike sorting workflow

Our spike sorting procedure has three main steps: data preprocessing, feature extraction, and clustering [52]. The innovation in our method lies in improving feature extraction by nonlinear dimensionality reduction of the spike waveforms and hierarchical clustering for classification as neural units.

### Data preprocessing

To minimize the noise within the waveforms, the third-order Savitzky–Golay filter was used, with a window of 5 samples. This filter effectively smooths the signal while preserving essential spike features. After filtering, waveforms are interpolated using a Piecewise Cubic Hermite Interpolating Polynomial, which maintains a smooth interpolation and preserves the original waveform shape. Each spike is then aligned at its minimum, followed by downsampling to return to the original point count while preserving alignment.

### Feature extraction

Following preprocessing, we applied UMAP [24] to achieve nonlinear dimensional reduction of the waveforms, using hyperparameters set as: n_components = 2, min_dist = 0, and n_neighbors = 15 [52]. To gauge performance in terms of

dimensionality, we established a grid that was purposed for hyperparameter tuning. This included searching the n_neighbors parameter while maintaining min_dist at 0, thereby ensuring maximum fidelity of density in the low-dimensional space. We then employed the Kolmogorov-Smirnov test to calculate the divergence between the normal distribution and the distribution of data in the low-dimensional space. The hyperparameters with the highest divergence were finally established. UMAP utilizes mathematical principles based on graph theory and differential geometry, thus allowing it to capture local and global arrangements of the data well. In comparison to other approaches, like t-SNE, UMAP offers improved performance in terms of global structure preservation, thus being particularly beneficial for large multidimensional data sets [24]. By maintaining nonlinear relationships between the data, UMAP offers a faithful low-dimensional representation of spike waveforms.

### Clustering

On the UMAP-reduced data, we utilized HDBSCAN [51] for clustering. HDBSCAN creates a dendrogram of clusters by point connectivity with the help of a minimum distance parameter [52]. Through condensation of the dendrogram, HDBSCAN identifies stable clusters, traveling down the hierarchy until an optimum density threshold is met, and selecting the most prominent clusters. Noise is natively handled in HDBSCAN by labeling points outside of dense regions as noise, and so it is an ideal algorithm to pair with UMAP for spike sorting. This low-dimensional subspace from UMAP also enables manual visualization and curation, allowing further validation by checking unit stability, ISI violations, and waveform consistency.

### Multielectrode array (MEA) analysis and spatial accuracy

We utilized a publicly available dataset [13,42] containing simultaneous intracellular (GT) and extracellular recordings from a 256-channel MEA to compare the performance of UMAP-based sorting and SC (Fig 5). SC was run using default parameters, processing all channels simultaneously (SC full MEA). For the UMAP pipeline, we employed three distinct approaches to assess robustness and the integration of spatial information:

1. **Single-channel UMAP:** UMAP and HDBSCAN were applied independently to the waveforms extracted from each electrode (Figs 5C, 5D, and S6A). The F1 score was calculated for the cluster best matching the GT neuron at each location. This approach assesses the robustness of the embedding under varying SNR conditions without leveraging spatial redundancy.

2. **UMAP multichannel analysis (geometric and simultaneous):** To leverage spatiotemporal information, we implemented strategies based on waveform concatenation. First, a reference "hotspot" electrode (the electrode closest to the GT neuron, typically showing the highest signal amplitude or F1 score in the single-channel analysis) was identified. Spike times detected on this hotspot electrode were used as the temporal reference. For each detected spike event, waveform snippets (61 samples; 30 pre- and 30 post-trough) were extracted from the hotspot and a specified set of neighboring electrodes at the corresponding time point. These snippets were concatenated end-to-end to form a single high-dimensional feature vector ("super-waveform"). This concatenated dataset was then processed using the UMAP and HDBSCAN pipeline.

   • **Geometric analysis (Cumulative):** We employed a cumulative approach to analyze the impact of progressively integrating spatial context. We defined concentric rings of neighboring electrodes centered on the hotspot (schema on Fig 5E). Starting with the hotspot, we iteratively incorporated electrodes from adjacent rings. In each step, the waveforms from the accumulated set of electrodes were concatenated, and the F1 score was calculated (Figs 5E, 5F, and S7A–S7F).

   • **Full MEA (Simultaneous):** In this approach, waveform snippets from all available electrodes across the MEA were concatenated simultaneously. UMAP and HDBSCAN were then applied to calculate a single, overall F1 score

representing the performance utilizing the entire array (see triangles in Fig 5C, 5D; and dashed lines in Figs 5E, 5F, and S7A–S7F). Also, average values for precision and recall were obtained by this method (see triangles in S8B and S8E Fig).

## Performance metrics and Inclusion Index Matrix

To compute the Inclusion Index Matrix $M_{i,j}$ (S2 Fig), we evaluated the percentage of spikes in raster plot $i$ that also appear in raster plot $j$ ($I_{i,j}$). $M_{i,j}$ is a directional index that quantifies the overlap between spike trains. As shown in Figs 6 and S2, this matrix is generally not symmetric.

In cases where spike times belong to the GT ($S_{GT}$), we establish the number of matched (True Positives, TP), missed (False Negatives, FN), and false-positive (FP) spikes with respect to a cluster $k$ ($S_k$), using a small time tolerance $\varepsilon$:

1. True Positives ($TP_k$): Spikes in $S_k$ matching $S_{GT}$ ($n_{match}^k = \left\{ \left| t_j^k - s_i \right| < \varepsilon \right\}$).

2. False Negatives ($FN_k$): Spikes in $S_{GT}$ not matched in $S_k$ ($n_{miss}^k := n_{GT} - n_{match}^k$).

3. False Positives ($FP_k$): Spikes in $S_{GT}$ not matching $S_{GT}$ ($n_{fp}^k := n_k - n_{match}^k$).

Using these definitions, we calculated the standard performance metrics:

- Precision $(P_k)$ : $P_k = \frac{TP_k}{TP_k + FP_k}$
- Recall $(R_k)$ : $R_k = \frac{TP_k}{TP_k + FN_k}$

If a GT neuron is available, we define the overall sorting performance using the F1 score:

- F1 Score $(F1_k)$ : $F1_k = \frac{2*(P_k*R_k)}{P_k + R_k}$

Note that precision and recall correspond to the GT column ($I_{k,\,GT}$) and the GT row ($I_{GT,\,k}$), respectively, in the inclusion matrices displayed in Fig 6. As shown in S2 Fig, low precision corresponds to high contamination, while low recall corresponds to high spike loss.

## Comparative analysis on different databases

The "Wave_Clus" [16] comprises 594 distinct average spike waveforms, derived from actual recordings, which served as templates for generating synthetic signals [40]. Randomly chosen spikes were added at arbitrary times and with varying amplitudes to replicate background noise. Simulation of various SNR conditions was accomplished by changing the ratio of the signal amplitude to noise amplitude. More details about the development of this database are given in reference [16,40].

We employed four large datasets in our study: C_Easy1, C_Easy2, C_Difficult1, and C_Difficult2. C_Easy1 had eight noise levels ranging from 0.05 to 0.4 with increments of 0.05, while the other three datasets included four noise levels ranging from 0.05 to 0.2.

As described in the original work, noise level is represented in terms of its standard deviation relative to the peak amplitude of the spikes. All spike classes had a peak value of 1. "Easy" and "difficult" describe the level of overlap of spikes in each dataset. To test the effectiveness of our method in complex scenarios, we used synthetic datasets simulating electrode drifting and bursty neuronal activity. Electrode drifting was modeled by progressively and linearly decreasing the amplitude of one spike class over time, from a value of 1.0 at the beginning of the recording to 0.3 at the end. Bursty activity was simulated using sequences of consecutive spikes with decaying amplitudes (e.g., 1.0, 0.7, and 0.5), in which action potentials were separated by an average of 3 ms (SD = 1, range: 1–5 ms) [16]. The results were then compared with the ones obtained from wavelet- and PCA-based methods. Spikes were detected from continuous recordings using

the ground-truth spike times available in the database. Each spike contained 64 sampling points, and the database was sampled at a frequency of 24 kHz.

In addition, the Buzsáki dataset [41,43] consists of the extracellular and juxtaposed intracellular recordings of 30 Sprague-Dawley anaesthetized rats. Electrical activity recordings were done at the hippocampus CA1 pyramidal layer.

For rodent hippocampal data [41,43] and synthetic data [16,40] (Figs 2 and 6), we compared UMAP with PCA and Wavelets, using the optimal number of PCA components and the optimal number of wavelet features. HDBSCAN clustering parameters were kept consistent across tests. The highest F1 score value cluster was used for comparison with the GT. For Fig 2, only spikes corresponding to a single synthetic neuron were treated as GT.

## Synthetic data simulations

We utilized the synthetic dataset [40] provided by Quian Quiroga and colleagues [16] to evaluate sorting performance under controlled conditions (Figs 2 and S3).

**Noise and low firing rates.** Background noise ($\eta$) is defined as the standard deviation of the background noise relative to the amplitude of the spike waveforms [16,40]. Following the methodology of [16], this noise is modeled as structured (colored) noise, generated by summing spike waveforms from non-sorted background unit at random times. To simulate low firing rates ("silent" neurons), we randomly subsampled one of the GT clusters.

**Bursting activity (amplitude variation).** To simulate the amplitude variations often observed during bursting activity (e.g., amplitude decrease during high-frequency firing), we modulated the amplitude of the spike waveforms within a cluster. The amplitude of the spikes was varied by multiplying the template waveform by a factor drawn from a uniform distribution ranging from 0.5 (50% reduction) to 1.0 (original amplitude).

**Waveform shape variation (Drifting simulation).** To simulate gradual changes in spike shape over time (often caused by electrode drift), we implemented a morphing procedure between two distinct GT templates (T1 and T2). The shape of spike $k$ in the sequence was generated as a linear interpolation: $S_k = (1 - \alpha_k)T1 + \alpha_k T2$. The interpolation factor $\alpha_k$ varied linearly from 0 to 1 across the sequence of spikes, simulating a gradual transition from T1 to T2 over the duration of the recording.

## Evaluating spike sorting performance

To assess the accuracy of different spike sorting algorithms, we used datasets with known GT and calculated performance metrics in terms of F1 score, precision, and recall (see S2 Fig).

## Spatial accuracy

For an in vitro mouse retinal MEA (256 channels), UMAP-based sorting was compared to SC (Fig 5), using database-specific parameters [13,42]. To assess spatial accuracy, the F1 score for each electrode was fitted to a decaying exponential as a function of distance ($x$) from the reference electrode (the electrode yielding the highest average F1 score among the 256 electrodes) as follows:

$$F1 = Ae^{-\frac{x}{\tau}} + B$$

where $\tau$ is the F1 score decay rate, and $A$ and $B$ parameters that adjust the exponential.

## Databases without ground truth

For the electrode recording database from rhesus monkeys (Figs 3 and 4), we applied identical data preprocessing to both UMAP and PCA feature extraction algorithms. For clustering, HDBSCAN was implemented (S1 Fig). We employed

recordings from the secondary somatosensory cortex (S2), the VPC, and the DPC during the interval comparison task [45–47] and the tactile detection task [44,46,59].

To compare the distributions of mean firing rates, we constructed cumulative distribution functions for the neurons sorted by UMAP and PCA within each recording area. The average firing rate was computed for each neuron during the foreperiod of both the interval comparison and tactile detection tasks. This firing rate was calculated across all interval trials, specifically during the pre-stimulus interval (see repository at [52]).

Spike data from different recording sessions during the foreperiod can be downloaded at [47].

Additionally, for each recording, we used several sessions to compare the number of neurons detected by each method and computed the corresponding mutual information values. The neural activity used for these calculations is available at [46], and the corresponding code can be found at [52].

## PCA sorting

The key point of performing PCA on spike waveform data for spike sorting is to discover an alternate coordinate system wherein the waveforms can be expressed in a more compact and reduced form. That is, the objective is to construct a low-dimensional subspace that contains most of the variance present in the high-dimensional waveform space. In general, there is a dramatic reduction in the number of significant dimensions, from a number equal to the number of sample points per waveform (e.g., several hundreds) to only a very small number of principal components (PCs) that account for the majority of the variance. PCA generates another coordinate system for the high-dimensional data, where the first PC accounts for the largest amount of variance relative to the shapes of the waveforms. The second PC accounts for the next largest portion of the variance, but each subsequent axis must be constrained to be orthogonal to all the previous axes.

PCA is derived from the covariance matrix of the waveform data, which is estimated across all waveforms and time points. The covariance matrix is provided by:

$$Cov\left(W_i,\,W_j\right) = \frac{\sum_{k=1}^{N}\left(w_i^k - \overline{w_i}\right)\left(w_j^k - \overline{w_j}\right)}{N-1} \qquad \text{(Eq. S1)}$$

where $N$ is the total number of waveforms (spikes) being considered, $w_i^k$ is the amplitude of $k$-th waveform at $i$-th time point. $\overline{w_j}$ is the mean amplitude at time point $j$ across all waveforms:

$$\overline{w_i} = \frac{1}{N}\sum_{k=1}^{N} w_i^k \qquad \text{(Eq. S2)}$$

The diagonalization of the covariance matrix $Cov\left(W_i,\,W_j\right)$, results in a new coordinate system represented by the columns of matrix V, and the columns are referred to as the derived axes or PCs. Λ is also a diagonal matrix with positive entries, where the diagonal entries of Λ represent the amount of variance in the waveform data captured by the corresponding PCs. Then we arrange the primary components based on how much variance they hold. The $k$-th PC projection of $l$-th waveform data is represented as:

$$\widehat{w_l} = \sum_{i=1}^{T} w_{li}v_{ik} \qquad \text{(Eq. S3)}$$

where $T$ is the number of time points in each waveform, $v_{ik}$ is the $i$-th element of the $k$-th PC, and $\widehat{w_l}$ is the projection of waveform $w_l$ onto $k$-th PC. Therefore, the PCs are linear combinations of amplitudes at different time points within the

waveform. The contribution of each time points to a given $\hat{w}_l$ is represented by the $i$-th element of the $k$-th column of eigenvectors matrix V. These PCs provide a low-dimensional description of the waveform data in this coding subspace, facilitating more effective spike sorting by highlighting the most significant variations in waveform shapes.

**Wavelet sorting**

The primary motivation behind the application of the wavelet transform in spike waveform data analysis for spike sorting is to obtain informative features that represent the waveforms in a more compact and concise way for later classification. The wavelet transform depicts each spike waveform as a collection of coefficients that embody both temporal and frequency details, thus enabling a close study of waveform configurations. By knowingly selecting a certain subset of important wavelet coefficients, we effectively decrease the data dimensionality from the original number of time points with which each waveform is described to a lower, more compact set of features that include the most relevant discriminative information.

The continuous wavelet transform (CWT), or discrete wavelet transform (DWT), is applied to each waveform. The wavelet transform of a waveform is defined as:

$$W(a, b) = \frac{1}{\sqrt{a}} \int x(t)\overline{\psi}\left(\frac{t-b}{a}\right) dt$$

(Eq. S4)

where $a > 0$ is the wavelet coefficient at scale. This coefficient represents the dilation, while $b$ represents the time shift or translation. Finally, $\overline{\psi}$, is the complex conjugate of the mother wavelet function $\psi$. In practice, we use the DWT due to computational efficiency and data discretization. The DWT decomposes the waveform into approximation and detail coefficients at various levels of decomposition. Mathematically, the DWT coefficients are computed using a series of high-pass and low-pass filters followed by downsampling.

Approximation coefficients at level $m$:

$$T_{mn} = \int_{-\infty}^{\infty} x(t)\psi_{mn}(t)dt$$

(Eq. S5)

Detail coefficients at level $j$:

$$S_{jn} = \int_{-\infty}^{\infty} x(t)\phi_{jn}(t)dt$$

(Eq. S6)

After decomposing the waveform into multiple levels, we select a subset of significant wavelet coefficients that capture the most relevant features for spike classification. This selection can be based on criteria such as the largest absolute values or statistical measures of variance across the dataset. The selected wavelet coefficients form a feature vector for each waveform:

$$d_m(t) = \sum_{n=-\infty}^{\infty} T_{mn}\psi_{mn}(t)$$

(Eq. S7)

where $m$ is the number of selected coefficients and $d_m$ are the detail coefficients at specific levels $m$ and positions $t$. These feature vectors provide a low-dimensional representation of the spike waveforms in a feature space suitable for clustering. Clustering algorithms (e.g., K-means, Gaussian Mixture) are then applied to the feature vectors to group similar spikes together, effectively sorting neurons based on the waveform shapes. Unlike PCA, wavelet decomposition does not

inherently rank its coefficients. To establish a ranking, we applied a previously proposed method [16]. Briefly, the Kolmogorov–Smirnov test was used to evaluate the distance between each coefficient's distribution and the normal distribution, with larger distances indicating greater divergence. The coefficients were then ranked accordingly, and the K coefficients with the highest values were selected for further analysis.

## Data analysis

**Firing rate.** For each neuron, a time-dependent firing rate was computed on a trial-by-trial basis using overlapping, rectangular, causal windows of 200 ms in length, with a step size of 20 ms (see code at [52]).

**Mutual information.** For each neuron, we quantified the relationship between the firing rate ($r(t)$) and the presented stimuli $s$ in each time window using Shannon mutual information [27,45,59]. This measure captures nonlinear dependencies between the two variables. To compute $I(r; s)$, we used the conditional firing rate distributions across stimulus conditions ($P(r|s)$) along with the overall firing rate distribution ($P(r)$), obtained by pooling trials across all interval values:

$$I(r;\,s) = \sum_i^N \sum_k^M P\left(r_i\middle|s_k\right) P\left(s_k\right) \log_2\left(\frac{P\left(r_i\middle|s_k\right)}{P\left(r_i\right)}\right)$$

(Eq. S8)

The mutual information significance was tested for each neuron for the given time intervals. We calculated a permuted mutual information value by conducting 1,000 permutations of the trials. A mutual information value was deemed significant if the probability that a permutation yielded an equal or higher value was lower than 0.05 ($p < 0.05$). Additionally, to account for the problem raised by finite sampling, we utilized the correction established in [4,45,59]. At the time of significance testing, we also used a multiple comparison correction by implementing a clustering method that has been described before [4]. This involved keeping only the group of time bins that had significant connectivity and a predefined size. For Figs 3 and S4, Shannon's information associated to time intervals ($I_{int}$), we considered the firing rate probability distribution associated with the different time intervals ($P(r|int)$) and the global firing rate distribution ($P(r)$) by combining trials from all intervals values. For Fig 4 we computed tactile information ($I_{tac}$) considering the firing rate probability distributions associated to threshold and suprathreshold tactile stimuli altogether ($P(r|tac)$) and the firing rate probability distributions in the absence of stimuli ($P(r|abs)$). The global firing rate distribution ($P(r)$) was computed by combining trials with the presence of tactile stimuli and the absence of them (see scripts at [52]).

## Supporting information

**S1 Fig. Comparison of three clustering methods— hierarchical density-based spatial clustering application with noise (HDBSCAN), K-means, and Gaussian Mixture—for neuronal spike classification.** This panel shows how each approach performs on four electrodes (A–D) during a time interval comparison task, where different colors denote the spikes assigned to putative neurons. HDBSCAN stands out for two key reasons: (1) it can reveal clusters with unusual shapes and densities (see electrodes A–D), and (2) it avoids forcing uncertain spikes (shown in pink) into any cluster. These unassigned points typically represent noise or non-neuronal signals, and excluding them sharpens the overall accuracy. In contrast, both K-means and Gaussian Mixture assign every spike to a cluster, which often misclassifies ambiguous events. Consequently, HDBSCAN more faithfully represents actual neuronal activity by filtering out only those questionable spikes that impair clarity, surpassing conventional feature-based clustering methods. The spikes recorded with the electrodes shown in the figure can be found at [46], and also at [52], along with the code to analyze them using the different clustering methods.
(PDF)

**S2 Fig. Schema of spike sorting performance metrics.** This figure illustrates how different sorting outcomes affect standard performance metrics (precision, recall, and F1 score) when comparing a sorted cluster ($S_i$) against a ground truth (GT) neuron $S_{GT}$. It also describes the Inclusion Index Matrix ($M_{i,j}$). **(A–D)** Venn diagrams illustrating different scenarios, along with the resulting 2 × 2 Inclusion Matrix visualization when one unit is GT (labeled $E_i$ in the matrix). Precision ($P_i = I_{i, GT}$) corresponds to element (1, 2), and recall ($R_i = I_{GT, i}$) corresponds to element (1, 2). **(A)** Unrelated/ Worst Case: Low precision and low recall (F1 ≈ 0). Few spikes are shared. **(B)** High Spike Loss: High precision and low recall. The cluster contains mostly GT spikes but misses many of them (False Negatives). **(C)** High Contamination: Low precision and high recall. Most GT spikes are captured, but the cluster includes many non-GT spikes (False Positives). **(D)** Optimal Sorting: High precision and high recall (F1 ≈ 1). The sorted cluster accurately represents the GT neuron. **(E)** Color Bar for the Inclusion Index values and structure of the Inclusion Index Matrix ($M_{i,j}$) for comparing two arbitrary spike trains $S_i$ and $S_j$.
(PDF)

**S3 Fig. Impact of noise, drifting, and low firing rates on Spike Sorting performance.** This figure illustrates how PCA-, Wavelet -, and UMAP-based methods respond to challenges frequently encountered in spike sorting: overlapping waveforms, background noise, electrode drifting, neuron bursting, and neurons with very low firing rates. All clustering was performed with HDBSCAN. **(A, B)** PCA (A) and Wavelet (B) projections for an example dataset from Quiroga and colleagues [16]. Colors indicate ground truth (GT) neuron identities, while each marker denotes a detected spike. Although adding higher dimensions can sometimes improve cluster separability, it also complicates visualization, and neither PCA nor Wavelet consistently isolates the three GT neurons. **(C)** Sorting performance (F1 score) under drifting (left) and bursting (right) conditions. UMAP maintains robust performance (blue trace), whereas PCA (orange trace) and Wavelet (green trace) degrade substantially when waveforms shift over time (drifting) or when neurons exhibit bursts of spikes. **(D)** Sample spike waveforms with added noise levels ($\eta$ = 0.05, 0.1, 0.15, 0.2) to synthetic data from Quian Quiroga and colleagues [16]. UMAP-based sorting is robust to increasing background noise. **(E)** Sorting performance (F1 score) as a function of projection dimensionality at different noise levels (left/middle/right panels for PCA, Wavelet, and UMAP, respectively). At high noise ($\eta$ = 0.2), UMAP still yields an F1 score close to 0.6 and generally outperforms both PCA and Wavelet. The synthetic data used to generate this figure are publicly available at [40], and the code for performing the analyses is available at [52].
(PDF)

**S4 Fig. Preserving low-firing-rate neurons with UMAP sorting in the Time Interval Comparison Task.** The figure shows how pooling spikes from several neurons into a single multiunit can eliminate important task-related encoding—particularly if there is a low-firing-rate neuron. **(A)** Cartoon showing how poor spike sorting artificially inflates firing rates by combining activity from different neurons, potentially obscuring their distinct encoding patterns. **(B)** Time Interval Comparison Task (TICT). Animals are comparing two stimulus intervals (Int1 and Int2), both 400–2000 ms, with 2-second gap. The animal must retain Int1 information through the gap and compare it with Int2. Recordings from dorsal premotor cortex (DPC). **(C)** Raster (top) and firing rate (middle) of a multiunit artificially created by pooling two neurons when using PCA-based sorting. Gray shows the stimulus period; black ticks indicate spikes during correct trials; red ticks indicate spikes during error trials. The mutual information trace (bottom) shows reduced encoding of Int1 since combined units lose valuable signals from low-firing units. **(D, E)** Single neurons sorted using UMAP that were conflated in (C). While the neuron in (D) fires rarely, however, it does manage to encode Int1 at the start of the delay phase—information lost in the multiunit representation (C). Conversely, the second neuron (E) has a diverging firing pattern and encodes unique dynamics of the task. These two neurons collectively convey far more information about Int1 than the conflated multiunit in (C). The multiunit and UMAP sorted neuronal activity used to generate the raster plots and firing rates is publicly available at [46], and the code to compute firing rates and mutual information is available at [52].
(PDF)

**S5 Fig. Preserving different encoding patterns with UMAP-based sorting in tactile detection task. (A)** Schematic showing how inadequate spike sorting inflates firing rates by combining separate neuronal activities, potentially masking each neuron's individual encoding. **(B)** Tactile detection task: Animals receive a vibrotactile stimulus (0–24 μm) on the fingertip and must indicate whether it was present or absent. Recordings are from the ventral premotor cortex (VPC, green). **(C)** Raster plots (top) and firing rate (bottom) for a multiunit formed by merging two neurons under PCA-based sorting. Trials are aligned to stimulus onset. Blue shading marks stimulus-present trials; gray shading marks stimulus-absent trials. Combining these spikes conceals each neuron's separate response profile, yielding an incomplete picture of their activities. **(D, E)** UMAP-sorted versions of the two underlying neurons from (C). The neuron in (E) maintains a prolonged response during the delay, while the neuron in (D) responds strongly but briefly. By separating them, UMAP-based sorting retains each neuron's individual activity, avoiding the loss of valuable low-firing-rate signals. The multiunit and UMAP sorted neuronal activity used to generate the raster plots and firing rates is publicly available at [46], and the code to compute firing rates and mutual information is available at [52].
(PDF)

**S6 Fig. Examples of UMAP spatial robustness in MEA recordings. (A)** Sorting performance (F1 score) versus distance for three example MEAs using single-channel UMAP sorting. **(B)** Spatial map of the F1 score across the MEA for the corresponding examples in A. Black squares indicate the region shown in C. **(C)** Waveform templates from the neuron with the highest F1 score, shown across the 5x5 electrode patch highlighted in (B). The largest amplitude waveform is recorded on the central electrode (closest to the GT), and the color saturation of each waveform corresponds to the local F1 score. UMAP robustly identifies the characteristic waveform across multiple neighboring electrodes. MEA recordings supporting the analyses are available at [42], and the code necessary for the analyses is available at [52].
(PDF)

**S7 Fig. Geometric analysis examples for multielectrode Uniform Manifold Approximation and Projection (UMAP) sorting. (A–F)** Six examples showing the decay of F1 score as a function of distance from the central electrode. The solid blue line (UMAP) tracks performance as signals from neighboring electrodes at progressively greater distances (d1→d6) are incorporated into the analysis. For comparison, dashed lines show the overall F1 score value when using the entire multielectrode array (MEA) signal at once for both UMAP (dashed blue) and SpyKING CIRCUS (SC, dashed light blue). UMAP consistently outperforms SC, particularly when leveraging this spatial information. MEA recordings supporting the analyses are available at [42], and the code necessary for the analyses is available at [52].
(PDF)

**S8 Fig. Influence of precision and recall on multielectrode array (MEA) sorting accuracy.** This figure examines how precision (related to contamination) and recall (related to lost spikes) contribute to the spatial decay of the F1 score (Fig 5) when comparing Uniform Manifold Approximation and Projection (UMAP) and SpyKING CIRCUS (SC) on MEA recordings. Results are averaged over multiple MEAs. **(A, D)** Average spatial matrices for precision (A) and recall (D) computed using UMAP-based sorting. **(B, C)** Decay of precision, and fit of that decay (C), as electrode distance increases for UMAP. Triangles on (B) represent average precision values computed with UMAP-based sorting (light-blue) or SC (cyan) using the entire MEA signal. **(E, F)** Decay of recall, and fit of that decay (F), as electrode distance increases. Triangles on (E) represent average recall values computed with UMAP-based sorting (orange) or SC (yellow) using the entire MEA signal. Note that average for recall using SC is significantly low with respect to UMAP's, indicating the loss of many GT spikes beyond the central electrode. By contrast, UMAP sustains higher precision and recall across a wider spatial range. MEA recordings supporting the analyses are available at [42], and the code necessary for the analyses is available at [52].
(PDF)

## Author contributions

**Conceptualization:** Alessio Franci, Román Rossi-Pool.

**Data curation:** Daniel Suárez-Barrera, Lucas Bayones, Norberto Encinas-Rodríguez, Sergio Parra.

**Formal analysis:** Daniel Suárez-Barrera, Lucas Bayones, Norberto Encinas-Rodríguez, Sergio Parra, Viktor Monroy, Sebastián Pujalte, Bernardo Andrade-Ortega, Román Rossi-Pool.

**Funding acquisition:** Román Rossi-Pool.

**Investigation:** Lucas Bayones, Héctor Díaz, Manuel Alvarez, Antonio Zainos, Alessio Franci, Román Rossi-Pool.

**Methodology:** Lucas Bayones, Norberto Encinas-Rodríguez, Bernardo Andrade-Ortega, Alessio Franci, Román Rossi-Pool.

**Software:** Daniel Suárez-Barrera, Lucas Bayones, Norberto Encinas-Rodríguez, Sebastián Pujalte, Román Rossi-Pool.

**Supervision:** Alessio Franci, Román Rossi-Pool.

**Validation:** Daniel Suárez-Barrera, Norberto Encinas-Rodríguez, Sebastián Pujalte, Alessio Franci.

**Visualization:** Daniel Suárez-Barrera, Lucas Bayones, Norberto Encinas-Rodríguez, Sergio Parra, Viktor Monroy, Sebastián Pujalte, Bernardo Andrade-Ortega, Alessio Franci.

**Writing – original draft:** Daniel Suárez-Barrera, Lucas Bayones, Alessio Franci, Román Rossi-Pool.

**Writing – review & editing:** Daniel Suárez-Barrera, Lucas Bayones, Norberto Encinas-Rodríguez, Alessio Franci, Román Rossi-Pool.

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
