## [Editor Report · Decision Letter 0]

11 Apr 2025

Dear Dr Rossi Pool,

Thank you for submitting your manuscript entitled "Relevance of Nonlinear Dimensionality Reduction for Efficient and Robust Spike Sorting" for consideration as a Research Article by PLOS Biology. Please accept my apologies for the delay in getting back to you. I am currently handling your manuscript on behalf of my colleague Christian Schnell since he is out of the office this week.

Your manuscript has now been evaluated by the PLOS Biology editorial staff and I am writing to let you know that we would like to send your submission out for external peer review.

IMPORTANT: After discussions within the team, we think that your manuscript would be a better fit as a 'Methods and Resources' article at the journal. Upon resubmission (see below), I would be grateful if you could please tick 'Methods and Resources' as the article type in the online submission form.

Before we can send your manuscript to reviewers, we need you to complete your submission by providing the metadata that is required for full assessment. To this end, please login to Editorial Manager where you will find the paper in the 'Submissions Needing Revisions' folder on your homepage. Please click 'Revise Submission' from the Action Links and complete all additional questions in the submission questionnaire.

Once your full submission is complete, your paper will undergo a series of checks in preparation for peer review. After your manuscript has passed the checks it will be sent out for review. To provide the metadata for your submission, please Login to Editorial Manager (https://www.editorialmanager.com/pbiology) within two working days, i.e. by Apr 13 2025 11:59PM.

Kind regards,

Richard

Richard Hodge, PhD

rhodge@plos.org

On behalf of:

Christian Schnell, PhD

cschnell@plos.org

PLOS

---

## [Decision Letter · Decision Letter 1]

17 Jun 2025

Dear Dr Rossi Pool,

Thank you for your patience while your manuscript "Relevance of Nonlinear Dimensionality Reduction for Efficient and Robust Spike Sorting" was peer-reviewed at PLOS Biology. It has now been evaluated by the PLOS Biology editors, an Academic Editor with relevant expertise, and by several independent reviewers.

In light of the reviews, which you will find at the end of this email, we would like to invite you to revise the work to thoroughly address the reviewers' reports.

As you will see below, the reviewers think that the your tool would be useful for the community. Both have a few concerns regarding the benchmarking and comparisons with existing tools, and the demonstration of the capabilities, which we think will need to be addressed in a revision.

Given the extent of revision needed, we cannot make a decision about publication until we have seen the revised manuscript and your response to the reviewers' comments. Your revised manuscript is likely to be sent for further evaluation by all or a subset of the reviewers.

**IMPORTANT - SUBMITTING YOUR REVISION**

*Re-submission Checklist*

*Published Peer Review*

*PLOS Data Policy*

*Blot and Gel Data Policy*

Sincerely,

Christian

Christian Schnell, PhD

Senior Editor

PLOS Biology

cschnell@plos.org

REVIEWS:

Reviewer #1 (Alessio Paolo Buccino): The authors of this manuscript developed a UMAP-based spike sorting pipeline and benchmarked its performance against other methods on simulated and experimental ground-truth data.

The manuscript also shows tha UMAP outperforms other dimensionality-reduction methods, especially in its ability to find low-firing units.

While the methodology is well described and sound, I think that there are some major points that need further work and clarification.

Major issues

1. Multi-electrode recordings:.

At several places, the authors claim that the method can improve spike sorting performance of high-density multi-electrode arrays. However, they present no results where the developed pipeline (and other methods) use the additional spatial information provided by multi-electrode probes.

In the last two sections of the results ("Comparing UMAP-based sorting and SpyKING CIRCUS in multielectrode array recordings: the role of distance" and "Comparing UMAP-based and Feature-based spike sorting in multi-electrode recordings"), the authors use multi-electrode data, but analyze the recording of each electrode separately.

This defeats the purpose of multi-electrode data, where the spatial redundancy is exactly what is used by different methods to refine the spike sorting. Spyking CIRCUS, for example, use data from multiple electrodes both for extracting features and for template matching. The analysis that they provide and the comparison with Spyking CIRCUS

is therefore limited and unfair in my opinion, and should be removed. Otherwise, the authors should let Spyking CIRCUS use all the available electrodes, since this is its intended use.

Similarly, the data from Buszaki is from tetrodes, that were developed primarily to increase the separability of neurons by recording them on multiple closed-by electrodes.

For these reasons, I think that the authors should either add a mutli-electrode implementation, e.g., where features from different electrodes are combined prior to clustering, or they should reframe the "multi-electrode" claims, and state that the method is primarily designed (as it stands) for single eletrode/microwire (and could be extended to multi-channel data).

I think that this could still be relevant, and the analysis of distance-related performance (in the comparison with Spyking CIRCUS) could further provide evidence on the fact that UMAP can work well on low SNR regimes, especially in relation to PCA and Wavelet methods.

2. Performance metrics.

The authors define additional metrics and names for measuring performance. Only in the methods, they explain that Igt,i is equaivalent to recall and Ii,gt is equivalent to precision. They further define an overall quality as the average between precision and recall.

I think that this new terminology is not needed and it makes the reading more confusing. Instead, I suggest to use existing metrics such as precision, recall throughout the manuscript. The quality score is not robust against imbalanced between precision and recall.

I believe that using more standard metrics, such as accuracy anf F1 score would be more familiar to readers and overall improve the clarity of the manuscript.

Minor issues

1, Title: Nonlinear Dimensionality Reduction can be any method. I suggest changing it to UMAP.

2. Introduction: silicon -> silicon probes

3. Introduction: Herding -> Herding Spikes

4. Introduction: different neurons) -> different neurons,

5. Introduction: These reasons is why -> These reasons are why

6. Introduction: "As the number of electrodes in an array increases, methods that require parameter hand-tuning for guaranteed performance become increasingly less usable." What methods require hand tuning? All the presented methods (PCA/Wavelet) are fully automatic

7. Intorduction: " using a 64-electrode multiarray throughout a week of recordings generates more than 300 different datasets, each consisting of a variable number of identifiable neurons". Whys is that? How did the authors come up with 300? One could record continuously for a full week...

8. Introduction: What do the authors mean with "expert-defined" metrics?

9. Results: "UMAP-based spike sorting vs. traditional feature-based pipelines" UMAP is also feature-baed. I would drop "tfeature-based" and just leave "traditional pipelines"

10. Results: ( ranging from 0.05 to 0.2) what are the units of this?

11. Results: " we randomly removed an increasing fraction of the datapoints from one of the synthetic GT neuron clusters" What is the original firing rate? How are spikes removed and what are the "final" firing rates?

12. Results: "thereby increasing the average number of neuron pairs from about 28 to almost 70." Why is this relevant? I tihnk it's enough to report the increased number of identified neurons.

13. Results: is ablet -> is able

14. Discussion: Reference 47 (SimSort) showed a deep-learning based method that outperforms UMAP. You should comment on that

15. Discussion / Methods: "Drifting" usually referes to the movement between electrodes/neurons, which causes a variation in spike shape over time. I would use something less misleading, like "varying dynamics"

16. Methods: How were these bursting and "drifting" simulations constructed? For bursting do you use a model of amplitude modulation vs spike index (as in https://pubmed.ncbi.nlm.nih.gov/32648042/)? What about drifting dynamics? What does it mean and how is it implemented?

17: Figure 3: what are the red spike trains?

18: Figure 6B: the spiketrains of different colors are hard to understand and interpret. How can a spike trains represent the "percentage of cluster spikes"?

19: Figure S2: letter D is missing

Reviewer #2: Summary and impression

In this work, the authors explore non-linear dimensionality reduction methods (specifically, UMAP) for the identification of single units on a single channel during extracellular electrophysiology. They compare this approach with two others that have been used in the past: PCA and wavelet decompositions and clustering methods like a GMM or K-means. They show that not only does UMAP equipped with HDBSCAN (a hierarchical clustering algorithm) better isolate the spikes arising from ground truth single units, they also are able to detect units with very low spike rates; this is something that traditional (density-based) methods fail to cluster. Furthermore, they demonstrate that erroneous isolation of units can confound downstream analyses that depend on assumptions of single unit isolation such as noise correlations and functional cell typing. This paper would be of interest to anyone building spike sorting tools.

Overall, I found the work to make an important and unifying contribution to the spike sorting literature. Specifically, while the utility of non-linear methods has been shown in a scattered way throughout the literature†, it has not been directly shown previously for cortical spikes obtained via threshold crossing and rigorously compared to extant approaches. While not a spike sorting tool, this research provides an important empirical result that I believe everyone working on spike sorting should be aware of.

Major Comments

I don't have any major questions or flaws in the work but I do think that there are a few clarifications/factual inaccuracies that should be included/amended. I view all the numbered major comments as being essential to at least address.

This work only is relevant for threshold-based methods. Non-linear methods with hierarchical clustering have also been shown to be highly effective for spike sorting on template-based approaches (Kilosort4). Please be more explicit about this as this result for template-based methods has technically been shown. However, having personally performed threshold- and template-based spike sorting, I believe the two settings are sufficiently different that the novelty of this approach is not in question. Furthermore, Kilosort4 provides an empirical result whereas this work also provides an answer as to "why" non-linear methods are generally superior. Still, I do believe that Pachitariu et al. deserves some mention for their non-linear approach even if they themselves did not make large mention of this change (from density-based methods) in their own work.

On the previous point, there are several factual inaccuracies that I believe should be fixed.

"Reducing the dimensionality of the processed data is mandatory to apply the clustering algorithms responsible for detecting, isolating and sorting the recorded putative neurons". Technically, this is not "mandatory" but it is however the predominant methodology. I think a phrase such as "near universally applied" or something would be better because it does not imply there is some hard requirement.

Factually untrue for spike sorting algorithms writ large because Kilosort4 does also apply clustering in high-dimensions as it uses a hierarchical clustering algorithm on the nearest neighbor graph.

What follows is more of a "statement rather than a question" that I don't expect the authors to incorporate in any way, it's just to make clear my perspective in case it is useful. To my knowledge, nothing fundamentally prevents clustering in ambient space. The good reason for why it is not done is that most methods use metrics inherited from the ambient space (Euclidean distance for example) to establish clusters and these metrics fail to elicit differences (and thus, don't yield meaningful clusters) because of concentration of measure in high-d. This is not just my perspective, it has been explicated by Leland McInnes himself (https://youtu.be/nq6iPZVUxZU?si=_wf554ZnWlYUUm3y&t=674). Graph clustering on a nearest-neighbor graph in high-dimensions (as KS4 does) tends to work very well. The reason why HDBSCAN on a low-d UMAP embedding also works very well is that UMAP too constructs a graph that is projected down into low-d in such a way as to optimize cluster structure over ambient global distance preservation whereas other methods like PCA do not (they preserve global distances).

"In all the aforementioned methods, the critical step . . .": So Kilosort4 is cited as one of the "aforementioned methods" but I would not describe it as being "ad-hoc nonlinear". It is no more ad hoc than the approach espoused here by the authors themselves.

"To the best of the authors' knowledge, all existing . . .": Well, KS4 and P-sort neither use "expert-defined, nonlinear projections". Also, there is a UMAP visualizer plugin for Phy2 so technically, that is another method.

An important point that is elided is that the crucial n_neighbors parameter has a large effect on the size and appearance of embeddings. In my experience, defaults tend to work well for most applications but this is frequently a point of contention levied by detractors of non-linear approaches. It should be provided some sense of what range of n_neighbor values to use and how HDBSCAN clusters are affected by it. I believe that there is some explanation of this in the "Feature extraction" section of the Methods but I do not remember this being mentioned in the main text.

Minor Comments

"Finally, powerful toolkits like KiloSort": the "S" in Kilosort should not be capitalized.

"Projection (P), in which the fuzzy simplicial complex is optimally embedded. . .": So technically an optimization procedure is followed (force-directed graph layout) but this is not "optimal" in any real sense. I would advise just removing the "optimally" from this sentence.

Figure 2E: in the legend there is "fr" listed—is this fraction of total firing rate? Some measure of the fraction of false positive spikes? This should be explained in the caption.

"Because noise is a critical factor . . .": What type of noise is this? Gaussian noise applied independently at each time point?

"while the (i+1)-th line, i = 1, …, 4, . . .": The index should end at "3" and not "4" since it's i+1 and there are only four total units shown.

"In spite of this, the utilization of the six most considerable coefficients (illustrated in Figs. 6I and 6J) . . .": I'm not sure how Figs. 6I and 6J make this point? Are the correct figures cited here?

Relative to other contemporary spike sorting approaches with UMAP, like WaveMAP": WaveMAP is not a spike sorting algorithm.

I'm not sure if this is a result of prior rounds of review of figure size/number limitations but I found that the supplementary figures so frequently referred to (and used to make important points about results) as to warrant them being main figures. I'm not strongly for or against the restructuring of this but I found it extremely hard to read as is.

†P-sort for cerebellum [Sedaghat-Nejad et al., 2021]; WaveMAP for cell type classification [Lee et al. 2023]; unnamed t-SNE method for cell type classification [Jia et al. 2019]; and nearest-neighbor graphs for template-based spike sorting [Pachitariu et al., 2024]

---

## [Decision Letter · Decision Letter 2]

22 Oct 2025

Dear Román,

Thank you for your patience while we considered your revised manuscript "Relevance of Nonlinear Dimensionality Reduction for Efficient and Robust Spike Sorting" for publication as a Methods and Resources at PLOS Biology. This revised version of your manuscript has been evaluated by the PLOS Biology editors, the Academic Editor and the original reviewers.

Based on the reviews and on our Academic Editor's assessment of your revision, we are likely to accept this manuscript for publication, provided you satisfactorily address the remaining points raised by the reviewers. Please also make sure to address the following data and other policy-related requests:

* We would like to suggest a different title to improve its accessibility for our broad audience:

"Efficient and reliable spike sorting from neural recordings with UMAP-based unsupervised nonlinear dimensionality reduction"

* Please add the links to the funding agencies in the Financial Disclosure statement in the manuscript details.

* DATA POLICY:

Regardless of the method selected, please ensure that you provide the individual numerical values that underlie the summary data displayed in the following figure panels as they are essential for readers to assess your analysis and to reproduce it: 4CDE.

CODE POLICY

We expect to receive your revised manuscript within two weeks.

*Published Peer Review History*

*Press*

Sincerely,

Christian

Christian Schnell, PhD

Senior Editor

cschnell@plos.org

PLOS Biology

Reviewer remarks:

Reviewer #1 (Alessio Paolo Buccino signed his report): The authors addressed all my issues.

I have just one minor comment left, regarding the clarification on "number of session":

"For example, assuming multiple acute sessions per day across a week—resulting

in, say, 5 sessions per electrode per week—the use of a 64‐electrode multiarray

generates approximately 320 (i.e., 64 electrodes recordings times 5 sessions)

independent extracellular activity datasets, each containing a variable and

unknown number of putative neurons, as a function of largely uncontrolled

session‐specific conditions."

I think that stating that each electrode is independent of each other is wrong. The authors now also incorporated a multi-electrode version and neurons appear on more that one electrode.

Honestly, I would just remove the entire sentenc: we all agree that with the increasing number of channels manual curation becomes unsuitable.

I thi

Reviewer #2: Overall, I am highly satisfied with the manuscript in both its structure and analyses and believe it will make a high-impact contribution to the field of electrophysiology. It crystallizes the notion that nonlinear methods are superior for clustering which is an essential operation in spike sorting. All of my substantive concerns have been fully addressed and I look forward to seeing this work in print upon which I will immediately it share with my colleagues.

I only had a few very minor edits to make but none are materially important. They only pertain to copyediting and terminology consistency issues especially in the Introduction which could use a closer read. I assume that these will all have been caught anyways in the proofing process but here they are anyways.

"To ensure efficient clustering of spiking waveforms, a necessary step in most spike sorting pipeline ...": pipeline should be plural.

"Template-matching methods like Kilosort4 (21, 22) have also recently demonstrated high performance by incorporating non-linear ...": non-linear should not be hyphenated for consistency (this also occurs a few other times throughout.

"UMAP algorithm (23, 24)": Missing the definite article "The" at the start of this sentence.

"and in efficiently identifying and classifying cell types (26-29). Besides, ...": Usage of the word "besides" is too colloquial in language in my opinion.

"properties of UMAP-based dimensionality reduction make it a perfect candidate ...": Again, the use of "perfect candidate" is probably too colloquial. Consider softening the claim here.

"By reducing data dimension, UMAP automatically identifies nonlinear geometric structures that serve as automatically-discovered geometric (nonlinear)": Nonlinear here is stated twice in the same sentence.

"Moreover, UMAP's robustness to data point density ensures that sparsely represented spike waveforms—such as those from neurons with low firing rates—can efficiently be identified and separated from densely represented waveforms originating from neurons with high firing rates.": I know what is meant here but the phrase "robustness to data point density" is ambiguous for a first-time reader and for its usage for the first time in the text. Consider something more specific such as "ability to capture waveform shape despite low firing rates" or some other explanation. The latter sentence "The proposed spike sorting method particularly stands out for its robustness to even large heterogeneities in the recorded neuron firing rates" is a good one.

"HDBSCAN is a clustering algorithm that does not require to pre-specify ...": should end with, "that does not require "the pre-specification of..."

"Although interesting, we do not explore this analogy further here.": It is highly unsatisfying to end the point here. I believe the authors but an uncharitable reader will not (and most readers are uncharitable).

"over which clustering is feasible, dimensionality-reduction...": Dimensionality reduction should not be hyphenated (this also occurs one other time in the text).

"F1 score: The harmonic mean of Precision and Recall.": Precision and recall should not be capitalized here or anywhere else in the text as they are not proper nouns.

"A high F1 score requires both Precision and Recall being high (the Optimal...": Optimal should not be capitalized.

"This capacity not only reduces manual curation but also ensures complete neuron detection, including silent or low-firing units typically omitted.": What is meant by "silent" is defined in the very next sentence and "silent" should be placed in quotation marks as the neurons are not actually silent but just low-firing rate.

"In contrast to methods like t-SNE or ISOMAP...": Isomap should only have the first letter capitalized.

---

## [Editor Report · Decision Letter 3]

12 Nov 2025

Dear Román,

Thank you for the submission of your revised Methods and Resources "Efficient and reliable spike sorting from neural recordings with UMAP-based unsupervised nonlinear dimensionality reduction" for publication in PLOS Biology. On behalf of my colleagues and the Academic Editor, Carl Petersen, I am pleased to say that we can in principle accept your manuscript for publication, provided you address any remaining formatting and reporting issues. These will be detailed in an email you should receive within 2-3 business days from our colleagues in the journal operations team; no action is required from you until then. Please note that we will not be able to formally accept your manuscript and schedule it for publication until you have completed any requested changes.

PRESS

Sincerely, 

Christian

Christian Schnell, PhD

Senior Editor

PLOS Biology

cschnell@plos.org